# Chickpea NCR13 disulfide cross-linking variants exhibit profound differences in antifungal activity and modes of action

James Godwin[1], Arnaud Thierry Djami-Tchatchou[1], Siva L. S. Velivelli[1], Meenakshi Tetorya[1], Raviraj Kalunke[1], Ambika Pokhrel[1], Mowei Zhou[2], Garry W. Buchko[2,3], Kirk J. Czymmek[1,4], Dilip M. Shah[1] *

1 Donald Danforth Plant Science Center, St. Louis, Missouri, United States of America, 2 Earth and Biological Sciences Directorate, Pacific Northwest National Laboratory, Richland, Washington, United States of America, 3 School of Molecular Biosciences, Washington State University, Pullman, Washington, United States of America, 4 Advanced Bioimaging Laboratory, Donald Danforth Plant Science Center, St. Louis, Missouri, United States of America

☯ These authors contributed equally to this work.
¤ Current address: Department of Chemistry, Zhejiang University, Hangzhou, Zhejiang, China
* dshah@danforthcenter.org

**Data Availability Statement:** The coordinates of the NCR13 NMR structure have been deposited in

## Abstract

Small cysteine-rich antifungal peptides with multi-site modes of action (MoA) have potential for development as biofungicides. In particular, legumes of the inverted repeat-lacking clade express a large family of nodule-specific cysteine-rich (NCR) peptides that orchestrate differentiation of nitrogen-fixing bacteria into bacteroids. These NCRs can form two or three intramolecular disulfide bonds and a subset of these peptides with high cationicity exhibits antifungal activity. However, the importance of intramolecular disulfide pairing and MoA against fungal pathogens for most of these plant peptides remains to be elucidated. Our study focused on a highly cationic chickpea NCR13, which has a net charge of +8 and contains six cysteines capable of forming three disulfide bonds. NCR13 expression in *Pichia pastoris* resulted in formation of two peptide folding variants, NCR13_PFV1 and NCR13_PFV2, that differed in the pairing of two out of three disulfide bonds despite having an identical amino acid sequence. The NMR structure of each PFV revealed a unique three-dimensional fold with the PFV1 structure being more compact but less dynamic. Surprisingly, PFV1 and PFV2 differed profoundly in the potency of antifungal activity against several fungal plant pathogens and their multi-faceted MoA. PFV1 showed significantly faster fungal cell-permeabilizing and cell entry capabilities as well as greater stability once inside the fungal cells. Additionally, PFV1 was more effective in binding fungal ribosomal RNA and inhibiting protein translation *in vitro*. Furthermore, when sprayed on pepper and tomato plants, PFV1 was more effective in reducing disease symptoms caused by *Botrytis cinerea*, causal agent of gray mold disease in fruits, vegetables, and flowers. In conclusion, our work highlights the significant impact of disulfide pairing on the antifungal activity and MoA of NCR13 and provides a structural framework for design of novel, potent antifungal peptides for agricultural use.

the Protein Data Bank, www.rcsb.org 8ULM (NCR13_ PFV1) and 7TH8 (NCR13_ PFV2) and the NMR chemical shifts have been deposited in the BioMagResBank, https://bmrb.io/ with accession codes 31111 (oxidized NCR13_PFV1) and 30979 (oxidized NCR13_PFV2).

**Funding:** The work presented here was supported by the National Science Foundation grant IOS-2037981 and the National Institute of Food and Agriculture GRANT13515963 to D.S and K.C. Salaries of D. S., C.Z. and G. J. were partly funded by the NIFA Grant 13515963 and NSF grant IOS-2037981. The funders had no role in study design, data collection and analysis, decision to publish, or preparation of the manuscript.

**Competing interests:** A.T.D.-T., S.L.S.V., M.T. and K.C. are affiliated with Invaio Sciences company. The other authors have declared that no competing interests exist.

## Author summary

Fungal pathogens cause significant pre-harvest and post-harvest losses of crop yield, making them a serious biological threat to global food security. Chemical fungicides are effective in controlling fungal diseases across various crops. However, rapid evolution of fungal pathogen resistance to single-site chemical fungicides in agriculture has created an urgent need for development of safe, sustainable, and cost-effective multi-site fungicides. Nodule-specific cysteine-rich (NCR) peptides expressed in the inverted repeat-lacking clade of legume plants exhibit potent antifungal activity; however, their modes of action (MoA) are poorly understood. Particularly, the specific contribution of disulfide pairing to the potency and spectrum of antifungal activity against fungal plant pathogens and MoA of these peptides remains to be identified. Chickpea NCR13 expressed in *P. pastoris* generates two peptide variants that differ in their disulfide cross-linking pattern. These variants exhibit striking differences in their three-dimensional structures and potency of antifungal activity against multiple fungal pathogens and MoA. They also differ in their ability to confer resistance to gray mold in pepper and tomato plants. Our study highlights the major impact a specific pattern of disulfide pairing can have on the *in vitro* and *in planta* antifungal activity of an NCR peptide.

## Introduction

The management of fungal diseases using chemical fungicides is one of the mainstays of modern agriculture. Unfortunately, modern fungicides that have a single site mode of action (MoA) are losing their effectiveness owing to an increase in development of fungal resistance [1]. A major limitation in fungicide development is finding molecules with novel structures and multi-site MoA to mitigate the evolution of resistance.

Plants express many antimicrobial peptides that are vital components of the innate immune system and act as a first line of defense against bacterial and fungal pathogens [2,3]. Moreover, plants express a multitude of cationic antifungal peptides differing in structure and sequence [4]. They are versatile, have large sequence space, and can be tuned for broad-spectrum antifungal activity against plant fungal pathogens. Thus, cationic antimicrobial peptides represent promising templates for a new generation of fungicides with broad action against fungal pathogens [5].

Several antifungal peptides expressed in plants are cysteine-rich with a characteristic number of cysteine residues at conserved positions and exhibit high degree of sequence diversity [4,6]. Defensins and defensin-like peptides are the best characterized examples of cysteine-rich peptides employed by plants to combat fungal pathogens [7–9]. Legumes of the inverted repeat-lacking clade (IRLC) express many nodule-specific cysteine-rich (NCR) defensin-like peptides that cause differentiation of nitrogen-fixing bacteria to bacteroids. In *Medicago truncatula* and chickpea, the terminal differentiation process is orchestrated by 639 and 63 nodule-specific cysteine-rich (NCR) peptides, respectively, with conserved cysteine motifs [10]. NCRs cause bacterial genome endoreduplication, changes in transcription and translation, and induction of morphological changes leading to the formation of bacteroids with unique morphology [8,11–13]. Each NCR is synthesized as a precursor protein containing an N-terminal signal peptide and a C-terminal mature peptide. Several NCRs in each legume are cationic, with a net charge ranging from +3 to +11, and rich in hydrophobic residues, dual hallmarks of defensin-like antimicrobial peptides [10,14]. Large landscape of natural NCRs from IRLC

legumes offer an unprecedented opportunity for discovery of the novel antimicrobial functions in plants. Several NCR peptides exhibiting antibacterial activity against Gram-negative and Gram-positive bacteria have so far been identified [15–18]. Another report detailed the evaluation of 19 cationic *M. truncatula* NCRs for their ability to inhibit the growth of a human fungal pathogen, *Candida albicans*. Of these, nine with a net charge above +9 inhibited the growth of this pathogen at concentrations ranging from 1.5 to 10.5 μM [14]. Previously, we reported *M. truncatula* NCR044 having broad-spectrum antifungal activity and multi-faceted MoA. In addition, when sprayed on leaf surfaces of tomato and *Nicotiana benthamiana* plants, NCR044 conferred resistance to the gray mold disease caused by the fungal pathogen *B. cinerea* [19].

NCRs exhibit diverse primary amino acid sequences and net charges, with a notable conserved feature being the presence of four or six conserved cysteine residues. These cysteines form intramolecular disulfide bonds which enhance the stability of the peptides [10,20]. The influence of cysteines and disulfide cross-linking on the structure and function of *M. truncatula* NCR247 and NCR169, each containing four cysteines, has been previously investigated. Four conserved cysteines and disulfide cross-linking had a major influence on the symbiotic activities of NCR247 tested against *Sinorhizobium meliloti* [21,22]. NCR169 produced two structurally distinct disulfide variants that exhibited differences in phospholipid binding and antibacterial activity [23]. However, the significance of disulfide cross-linking for the structures, antifungal activity, and MoA of majority of the NCRs is poorly understood.

In this study, we show that a highly cationic chickpea NCR13 peptide expressed in *Pichia pastoris* forms two structurally unique disulfide variants despite having an identical amino acid sequence. Interestingly, these two variants exhibited distinct differences in their antifungal potency against economically important plant fungal pathogens. NCR13 disulfide variants displayed notable differences in timing of fungal cell entry and plasma membrane permeabilization, stability, and MoA against the fungal pathogen *B. cinerea*. While both disulfide variants reduced gray mold disease symptoms by spray application on the leaves of pepper and tomato plants, the NCR13 disulfide variant with potent *in vitro* antifungal activity was more effective in controlling the disease *in planta*. Our study reveals the remarkable impact of disulfide cross-linking on the structure-activity relationships and MoA of an antifungal defensin-like NCR peptide.

## Results

### Chickpea NCR13 is expressed in *P. pastoris* as two peptide folding variants differing in disulfide pairing

In chickpea, NCR13 is one of 63 predicted NCR peptides expressed only in nodules undergoing differentiation during symbiosis [10,24]. NCR13 is a 32-residue peptide with six cysteines predicted to form three disulfide bonds (Fig 1A). Two other homologs of this peptide, NCR07 and NCR15, are also expressed in chickpea nodules with highly conserved cysteine residues (Fig 1A). No other IRLC legumes express NCRs with significant homology to NCR13. NCR13 has 66.6% and 48.4% sequence identity with NCR15 and NCR07, respectively, carrying a net charge of +8 and a hydrophobic amino acid content of 28%, hallmark features of a potential antifungal peptide (Fig 1B).

To investigate the antifungal properties of the NCR13 peptide, it was expressed in a heterologous *Pichia pastoris* system. The peptide was secreted into the growth medium and purified using cation exchange chromatography. When the NCR13 peptide was subjected to C18 reverse-phase high-performance liquid chromatography (HPLC), we consistently observed two peaks (Fig 1C). In contrast, chemically synthesized NCR13 (NCR13_CS) produced only one peak (Fig 1D). These observations led us to suspect that oxidized NCR13 was expressed in

A

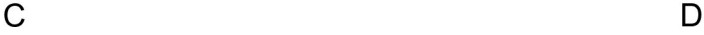

```
CaNCR13  -TKPCQSDKDCKKFACRKPKVPKCINGFCKCVR-  32
CaNCR15   TKQPCKSRKHCKTYRCPTPKVPNCVNGFCKCVR-  33
CaNCR07   KKMPCKRRRDCKTYPCPHPKVRDCVKGYCKCVVR  34
```

B

| Peptide | Net Charge | % Hydrophobic amino acids | Predicted MW (kDa) | Isoelectric point | % identity with NCR13 |
|---|---|---|---|---|---|
| CaNCR13 | +8 | 28.1 | 3.66 | 9.59 | 100 |
| CaNCR15 | +9 | 24.2 | 3.81 | 9.82 | 66.6 |
| CaNCR07 | +10 | 26.5 | 4.08 | 9.85 | 48.4 |

C

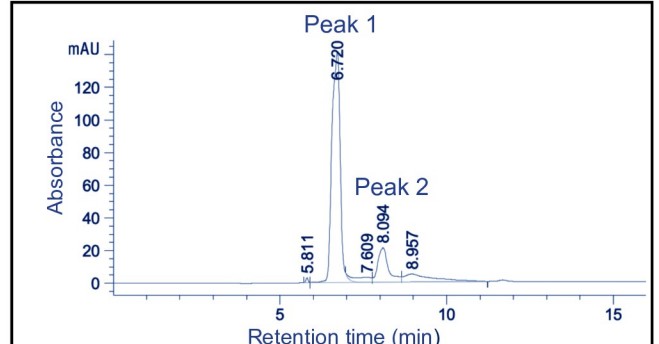

D

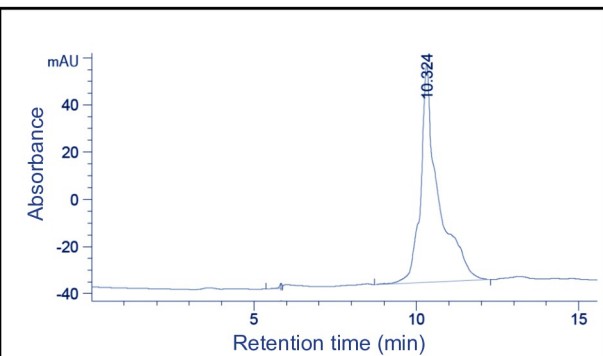

**Fig 1. Primary amino acid sequence and properties of NCR13.** (A) Primary amino acid sequences for Chickpea *(C. arientinum)* NCR13 and its homologs NCR15 and NCR07. Conserved cysteine residues are highlighted in red. The number of amino acids is indicated at the end of each sequence. (B) Peptide characteristics of CaNCR13, CaNCR15 and CaNCR07. (C) High-performance liquid chromatography (HPLC) analyses of NCR13 produced recombinantly in *Pichia pastoris*, monitored at 260 nm, reveals the presence of two major products. mAU represents milli-Absorbing Units. (D) Chemically synthesized NCR13 (NCR13_CS) elutes as one HPLC peak.

our *P. pastoris* expression system as two species differing in length or disulfide cross-linking pattern. Mass spectrometry analysis of peptides purified from both HPLC peaks revealed identical molecular weights (S1 Fig) expected for NCR13 with three disulfide bonds, narrowing our suspicion to different disulfide bonding patterns. Across various batches of purified NCR13, peak 1 was consistently observed to contain less peptide than peak 2 (S1 Table). Peak 1 and peak 2 peptides are hereafter referred to as NCR13_Peptide Folding Variant 1 (NCR13_PFV1) and NCR13_PFV2, respectively.

To further characterize both NCR13_PFVs, we performed NMR structural analysis of [15]N-labeled NCR13_PFV1 and NCR13_PFV2. Fig 2A and 2B show the assigned [1]H-[15]N HSQC spectra for NCR13_PFV1 and NCR13_PFV2, respectively. The broad dispersion of amide chemical shifts in both spectral dimensions is a standard feature of a structured peptide [25]. As illustrated in Fig 2, assignment of both spectra was consistent with the primary amino acid sequence of NCR13. However, the pattern of the fingerprint [1]H-[15]N HSQC spectrum for both

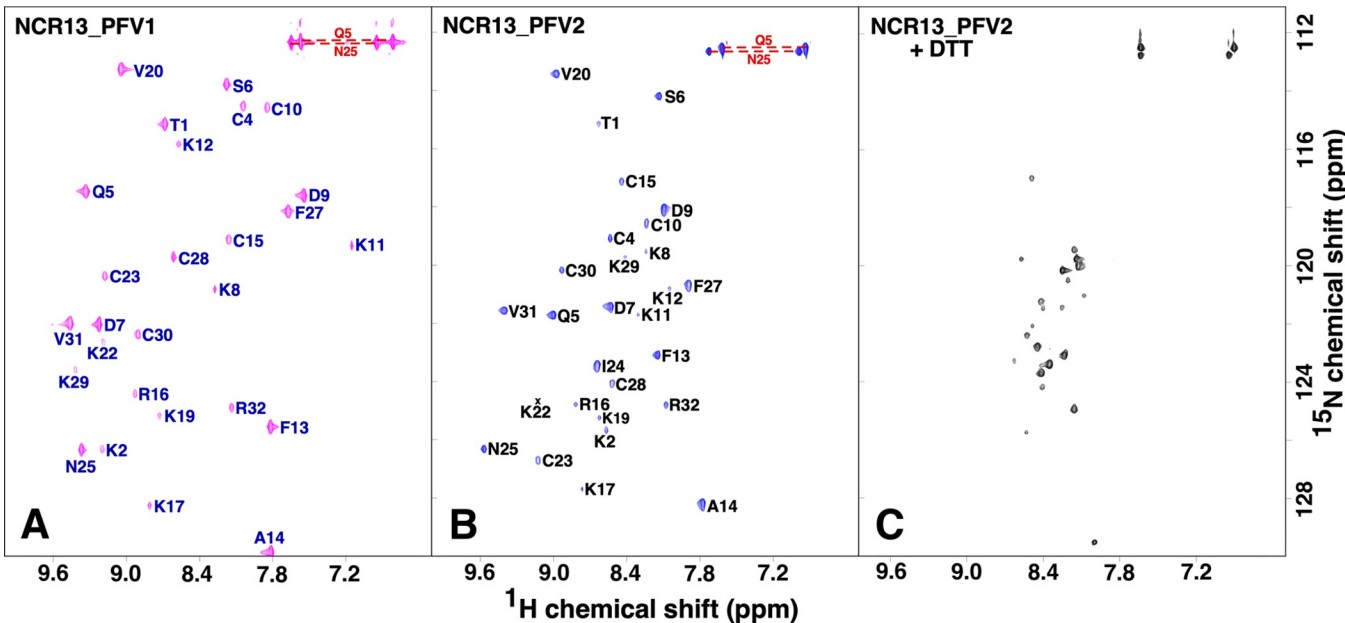

**Fig 2. Fingerprint $^1$H-$^{15}$N HSQC spectra for oxidized and reduced NCR13.** Assigned $^1$H-$^{15}$N HSQC spectra for oxidized (disulfide) NCR13_PFV1 (A) and NCR13_PFV2 (B). (C) The $^1$H-$^{15}$N HSQC spectrum of fully reduced (dithiol) NCR13_PFV1 and NCR13_PFV2 were identical (the spectrum for NCR13_PFV2 is illustrated). All spectra collected on ~ 1 mM samples at 20°C in 20 mM sodium acetate, 50 mM NaCl, pH 5.3 at a $^1$H resonance frequency of 600 MHz. Amide side chain resonance pairs are connected by a red dashed line. Not within the displayed spectral boundaries is the G26 amide resonance for NCR13_PFV1 and NCR13_PFV2 at 8.66/103.1 ppm and 7.99/101.7 ppm, respectively, and the I24 amide resonance for NCR13_PFV1 at 8.72/132.0 ppm.

peptides was clearly different, suggesting their structures were unique. Given that the primary amino acid sequence of NCR13 contains six cysteine residues (C4, C10, C15, C23, C28, and C30) that can form fifteen different patterns of intramolecular disulfide bonds when all six are oxidized, our attention immediately focused on a different disulfide bond pattern for the two peptides. The treatment of both peptides with the reducing agents dithiothreitol (DTT) or tris (2-carboxyethyl) phosphine resulted in the identical $^1$H-$^{15}$N HSQC spectrum shown in Fig 2C, illustrating that the parent primary amino sequence was identical for both peptides. Note that when the NCR13 cysteines are all in the reduced (dithiol) state, dispersion of amide chemical shifts in both spectral dimensions were distinctively narrow, a standard feature of an unstructured protein [25,26]. As observed for *M. truncatula* NCR044 [19], disulfide bond formation is essential to stabilize the elements of secondary structure observed in the oxidized state because in the reduced state the peptide is largely disordered.

Disulfide bond formation between all six cysteine residues in NCR13_PFV1 and NCR13_PFV2 was unambiguously verified from the NMR chemical shifts of the cysteine side chain β-carbons. In general, the cysteine $^{13}$C$^β$ chemical shift is greater than 35 ppm in the oxidized state and under 32 ppm in the reduced state. As summarized in S2 Table, the $^{13}$C$^β$ chemical shift for all six cysteines in both isoforms of oxidized NCR13 are between 37 and 48 ppm, indicating they are all oxidized in NCR13_PFV1 and NCR13_PFV2. Due to the number of potential disulfide bond patterns that the six cysteine residues can adopt, it was not possible to use NMR structures in early calculations to unambiguously deduce disulfide bond pairs from $^1$H-$^1$H NOEs that arise due to proximal γ-sulphur atoms in the disulfide state. Instead, as previously applied to NCR044, the disulfide bond pattern was inferred from the analysis of mass spectral data for trypsin-digested, unlabeled protein in the oxidized and reduced states [19]. For NCR13_PFV2, this approach showed that the disulfide bonds were C4-C10, C15-C30, and

C23-C28 (S2 Fig). For NCR13_PFV1, despite extensive modification of the experimental procedures, this approach failed due to the hyperstability of this isoform towards trypsinization. Consequently, the disulfide bond pattern predicted by AlphaFold [27] C4-C23, C10-C28, and C15-C30, was used to calculate the ensemble of NMR structures for NCR13_PFV1. The two NCR13 isoforms share one identical disulfide bond, C15-C30, and S2 Table shows that these cysteine $^{13}C^\beta$ chemical shifts differ less than 0.5 ppm, strongly suggesting this disulfide bond is present in both isoforms. As will be discussed below, this disulfide bond pattern best supports the experimental NOE data.

## Three-dimensional NMR solution structures of NCR13_PFV1 and NCR13_PFV2

To obtain structural insights into NCR13_PFVs, we performed NMR analyses of the oxidized forms of these two peptides. Fig 3 illustrates a superposition of cartoon representations of the final ensemble of structures calculated for the two NCR13 PFVs (left) with the main chain of a single structure shown to the right of the ensemble with the disulfide bonds highlighted in different colors. The elements of secondary structure are essentially identical in both isoforms, one anti-parallel β-sheet (V20-I24, F27-V31) and a short α-helix (Q5 or S6–C10). However, the disulfide bond pattern markedly alters the relative orientation of these elements of secondary structure. Both peptides contain one identical disulfide bond, C15-C30, that ties the N-terminal end of the peptide to approximately the middle of the linker region between the two elements of secondary structure. The consequence is that in NCR13_PFV1 the α-helix sits over the top of the face of the β-sheet while in NCR13_PFV2 the α-helix sits on the side of the β-sheet. This difference is better illustrated in S3 Fig, a cartoon illustration of two representative structures superimposed on the β-strand. This results in a smaller accessible surface area for NCR13_PFV1 (3188 ± 63 Å²) than for NCR13_PFV2 (3357 ± 129 Å²). The more compact structure for NCR13_PFV1, with the disulfide bonds formed between C4, C10, C23, and C28 directed into the interior of the peptide, may protect these disulfide bonds from reduction and contribute to the hyperstability of NCR13_PFV1 towards reduction (significantly longer incubation times were required to reduce NCR13_PFV1 relative to NCR13_PFV2). Disulfide bonds may also be protected from reduction because this NCR13 isomer is less dynamic. This suggestion is indirectly made by comparing the ensemble of calculated solution structures illustrated in Fig 3. The ensemble of structures calculated for NCR13_PFV1 converge to a greater extent than for NCR13_PFV2 with the backbone (N-C$^\alpha$-C = O) RMSD to the mean for the ordered residues almost twice as large for NCR13_PFV2 (0.41 ± 0.13 Å versus 0.76 ± 0.20 Å, respectively; S3 Table). While this difference reflects the greater number of NOEs observed for NCR13_PFV1 (316) than for NCR13_PFV2 (119) (S3 Table) due to a more compact structure, it also indirectly suggests the NCR13_PFV1 explores less conformational space than NCR13_PFV2. In both isomers, the organization of the elements of secondary structure do not appear to be stabilized by a hydrophobic core as the hydrophobic residues are mostly solvent exposed. Consequently, the overall structure of each peptide is largely governed by the β-sheet and three disulfide bonds. It is not clear how important this observation is with regard to the antifungal activity of these peptides.

   To further verify that the AlphaFold predicted disulfide bond pattern was correct for NCR13_PFV1, an NMR structure was calculated using only the NOE and TALOS+ (dihedral angle) data (no hydrogen bonds) with the other possible disulfide bond pattern (C4-C28, C10-C23, and C15-C30). In S4A Fig, two representative cartoon structures that result from the two possible disulfide bond patterns are superimposed on the β-strands. In the reversed, C4-C28/C10-C23 disulfide pattern (yellow), the direction of α1 is also reversed relative to the

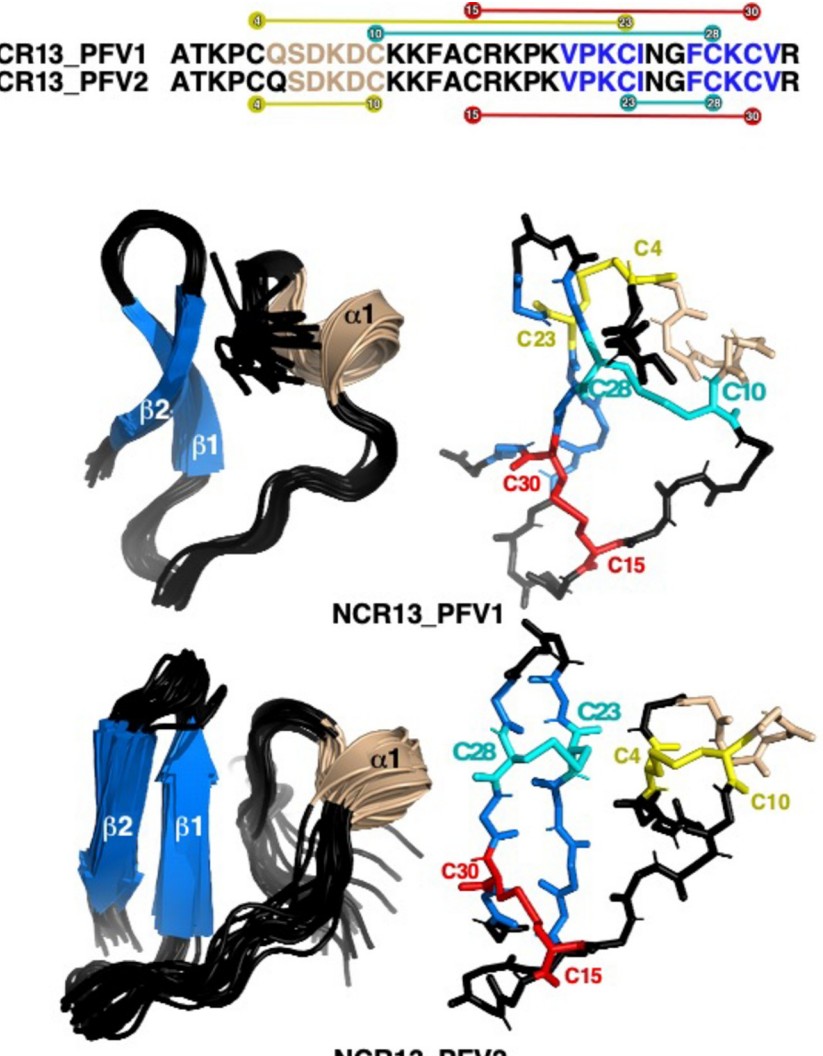

**Fig 3. Three-Dimensional NMR solution structures of NCR13_PFV1 and NCR13_PFV2.** Cartoon representation of the backbone superposition of the ordered regions in the ensemble of 20 structures calculated for oxidized NCR13_PFV1 (8ULM) and NCR13_PFV2 (7TH8). β-strands are colored blue and labeled sequentially starting from the N-terminal and the lone α-helix is colored wheat. Next to each ensemble is a backbone stick representation of a single structure with the six oxidized cysteine side chains highlighted in yellow, red, or cyan. Above the structures is a schematic summary of the elements of secondary structure observed for the two NMR structures, color-coded similarly. Also shown in the schematic are the disulfide bond connections in both isomers. One disulfide bond, between C15 and C30 (red), is identical in both isomers with C10 and C28 swapping disulfide bond partners.

β-strand and this prevents the N-terminal region (T1-P3) from making close contact with the β-strand. While most of the NOEs were satisfied in both structure calculations, long range experimental NOEs between the N-terminal region of NCR13_PFV1 (T1-P3) and the β-strand were only satisfied with the C4-C23/C10-C28 disulfide pattern used in the structure deposited to the PDB (8UTM), suggesting this pattern was correct. Furthermore, as shown in S4B Fig, the predicted AlphaFold structure for NCR13_PFV1 was like the experimental structure (8UTM). In both structures, the β-sheet is almost identical with the major difference a slightly shorter α-helix displaced two residues (D7-K11 versus Q5-C10 experimentally) that packs against the β-strand a bit differently in the AlphaFold structure (green). Most relatively, the N-

terminal region (T1-P3) folds towards the β-strand in the AlphaFold predicted structure, as verified experimentally with the long-range NOEs between these regions.

## NCR13_PFV1 exhibits much greater antifungal activity against fungal pathogens than NCR13_PFV2

Having found striking differences in their NMR structures, we investigated the *in vitro* antifungal activity of NCR13_PFV1 and NCR13_PFV2 against three necrotrophic plant fungal pathogens [19]. NCR13_PFV1 inhibited the growth of *B. cinerea* at the Minimal Inhibitory Concentration (MIC) of 0.09 μM. Conversely, NCR13_PFV2 inhibited *B. cinerea* at the MIC of 1.5 μM (Fig 4A–4C), seventeen-fold higher than that of NCR13_PFV1. NCR13_CS had a MIC of 3 μM, two-fold higher than NCR13_PFV2 (Fig 4B), suggesting that this fully reduced synthesized product was yet another isoform of NCR13 and illustrating the importance of disulfide bonds for antifungal activity. Similarly, NCR13_PFV1 showed several fold greater *in vitro* antifungal activity against *Fusarium virguliforme*, and *Sclerotinia sclerotiorum* compared to NCR13_PFV2 (Fig 4A). Overall, these results reveal that NCR13_PFV1 is a much more potent antifungal peptide compared to NCR13_PFV2.

## Disulfide bonds in NCR13_PFV1 contribute to the antifungal activity and formation of peptide folding variants

Fully oxidized NCR13_PFV1 contains three disulfide bonds: C4-C23, C10-C28, and C15-C30. The potent antifungal activity of NCR13_PFV1 against *B. cinerea* at nanomolar concentrations

**A**

| Pathogen | NCR13_PFV1 | NCR13_PFV2 |
|---|---|---|
| | MIC (μM) | |
| *Botrytis cinerea* | 0.09 | 1.5 |
| *Fusarium virguliforme* | 0.09 | 1.5 |
| *Sclerotinia sclerotiorum* | 6.0 | 24 |

**B**

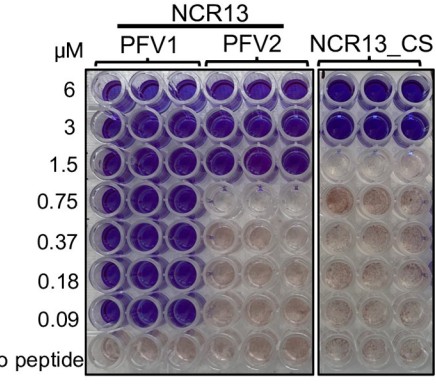

**C**

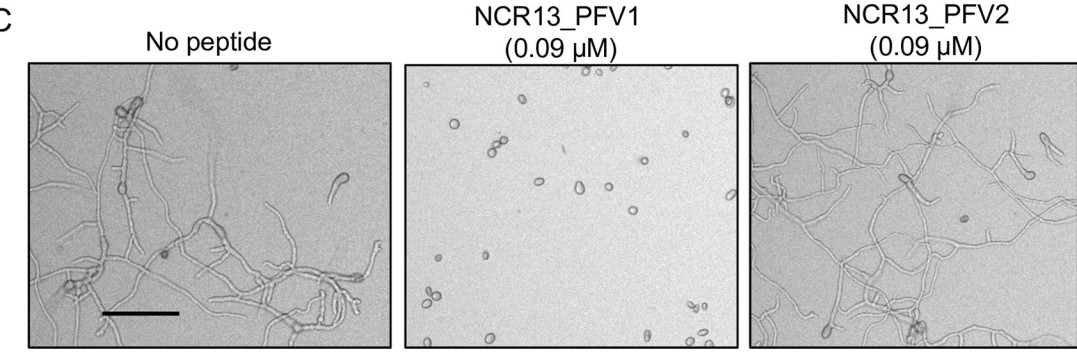

**Fig 4. NCR13_PFV1 exhibits potent antifungal activity compared to NCR13_PFV2.** (A) Minimal Inhibitory Concentration (MIC) values of NCR13_PFVs for different fungal pathogens. (B) Antifungal activity of NCR13_PFV1, NCR13_PFV2, and reduced NCR13_CS against *B. cinerea*. Fungal cell viability assay using resazurin dye. Change from blue to pink/colorless signals resazurin reduction and indicates metabolically active *B. cinerea* germlings after 60 h. NCR13_PFVs are used at concentrations of 0.09–6 μM, N = 3, N indicates biological replicates. (C) Representative microscopic images showing the inhibition of *B. cinerea* growth 24 h after treatment with 0.09 μM NCR13_PFVs (Right). *B. cinerea* without peptide added served as a negative control (Left). Scale bar = 100 μm.

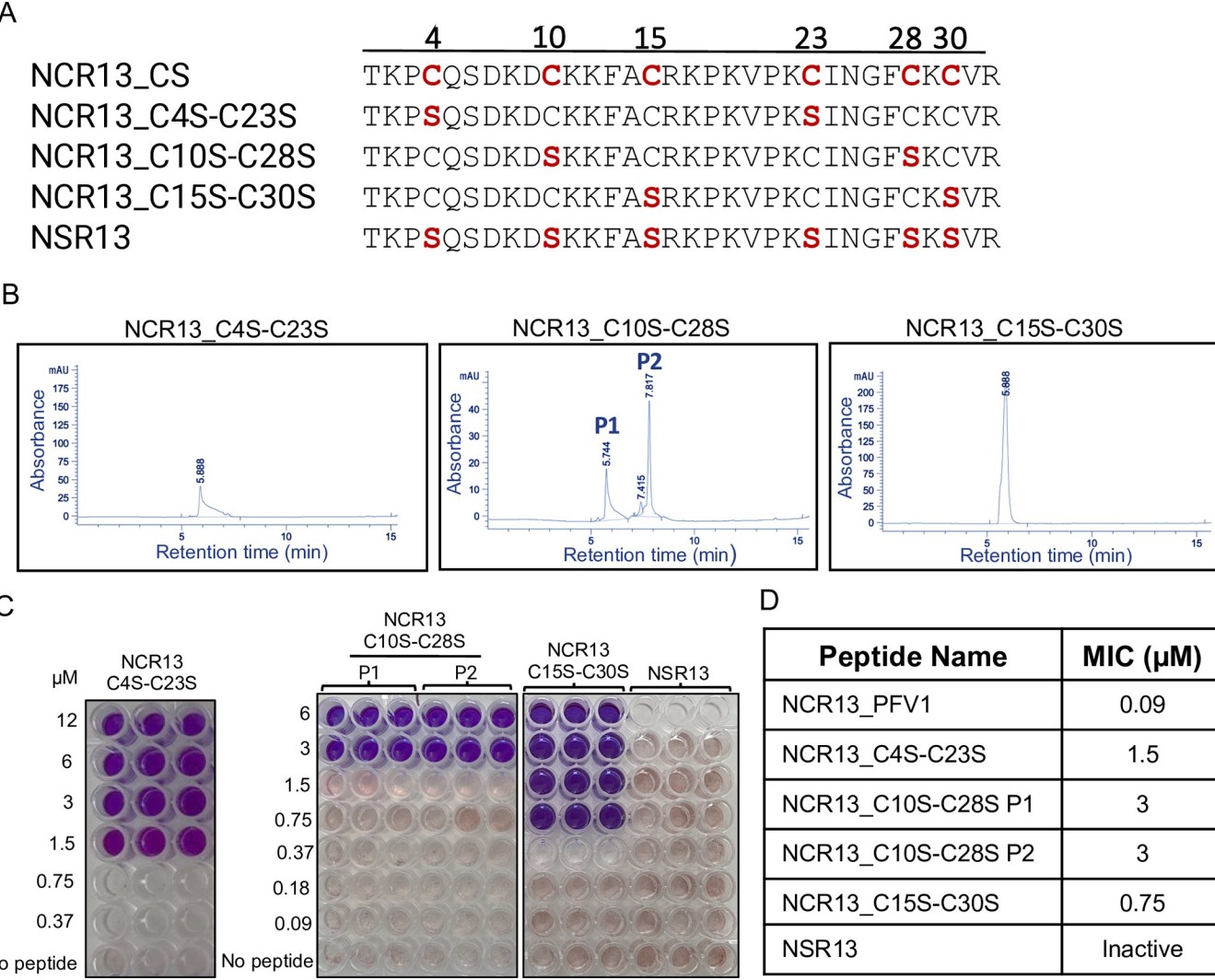

**Fig 5. Removing disulfide bonding in NCR13_PFV1 reduces antifungal activity against *B. cinerea in vitro*.** (A) Four NCR13 constructs used to assay the importance of disulfide bond formation for antifungal activity by substituting pairs of Cys residues with Ser (highlighted in red). The numbers above the sequence indicate the position of the Cys substitution. (B) HPLC analyses of NCR13_C4S-C23S, NCR13_C10S-C28S, and NCR13_C15S-C30S recombinantly produced in *P. pastoris* monitored at 260 nm (mAU = milli-Absorbing Units). (C) Fungal cell viability assay of disulfide knockout variants using the resazurin assay. A color changes from blue to pink/clear indicates of metabolically active (live) *B. cinerea* germlings after 60 h. (D) Summary of the MIC values for NCR13_PFV1 and NCR13 disulfide knockout variants.

prompted us to investigate the contribution of each disulfide bond towards antifungal activity and the formation of folded variants when expressed in *P. pastoris*. We carried out functional analysis of these disulfide bonds by creating three dual Cys to Ser variants to systematically remove one disulfide bond: NCR13_C4S-C23S, NCR13_C10S-C28S, and NCR13_C15S-C30S variants (Fig 5A). These knockout variants were expressed in *P. pastoris* and purified. HPLC profiles of NCR13_C4S-C23S, and NCR13_C15S-C30S revealed a single peak whereas NCR13_C10S-C28S revealed two peaks (Fig 5B), the latter observation suggesting NCR13_C10S-C28S folded into two presumed disulfide cross-linking variants each containing two disulfide bonds. However, the disulfide pairing pattern of the four Cys residues in these variants remains to be determined. Mass spectrometry analysis confirmed the expected molecular mass of each peptide (S4 Table), consistent with them being in an oxidized state.

Therefore, our results indicate that the disulfide pairing of C4-C23 and C15-C30 plays a crucial role in the formation of distinct NCR13_PFVs. Nevertheless, we recognize that different alternate disulfide arrangements can also yield functional forms, as observed with NCR13_PFV2.

All three single disulfide bond knockouts had lower antifungal activity against *B. cinerea* compared to NCR13_PFV1, as summarized in Fig 5. NCR13_C4S-C23S and NCR13_C15S-C30S both inhibited *B. cinerea* with a MIC value of 1.5 and 0.75 μM respectively, while the products in the two NCR13_C10S-C28S HPLC peaks had a MIC of 3 μM (Fig 5C and 5D). While we were unable to confirm the disulfide bonding pattern in each of these knockout variants, the MIC data indicates that each disulfide bond contributes to the antifungal activity of NCR13_PFV1 (Fig 5C and 5D). Indeed, an NCR13 construct with all six Cys residues replaced with Ser (NSR13) was inactive against *B. cinerea* even at the highest tested concentration, 24 μM (Figs 5C, 5D and S4). It is likely that NSR13 is unfolded, as suggested by the NMR data for fully reduced NCR13 (Fig 2) and the late elution time for NCR13_CS (Fig 1D) on a reverse phase column. Therefore, it may be concluded that Cys residues are essential for NCR13's antifungal activity with disulfide bonds essential for realizing its full antifungal potential.

## Identification of the sequence motif governing the antifungal activity of NCR13

To identify the primary amino acid sequence motif governing the antifungal activity of NCR13 against *B. cinerea*, we purchased chemically synthesized alanine scanning variants NCR13_AlaV1 through NCR13_Ala5 (Fig 6A). These variants contained a string of three to five consecutive alanine (Ala) residues substituted throughout the NCR13 sequence with altered net charge of the parent primary amino acid sequence. We then compared their antifungal activity relative to NCR13_CS. All experiments were conducted in the reduced state of the peptides lacking disulfide bonds. Notably, our findings revealed that NCR13_AlaV2, NCR13_AlaV3, and NCR13_AlaV4 had strong reduction in the net charge and no antifungal activity against *B. cinerea* at the highest tested concentration, 12 μM (Figs 6A and S6A), whereas NCR13_CS inhibited this pathogen at an MIC of 3 μM. This data suggested that a 13-residue, positively charged Lys-rich motif, KKFA**C**RKPKVPK**C,** located in the random coil and β-strand region, was a major determinant of NCR13 antifungal activity. We further explored the importance of this motif by testing antifungal activity of NCR13_AlaV3 expressed in *P. pastoris* to allow formation of disulfide bonds and found that it also lacked antifungal activity against *B. cinerea* like the chemically synthesized NCR13_AlaV3 (S6B Fig). It is noteworthy that NCR13_AlaV1 and NCR13_AlaV5 also displayed 2- and 4-fold reduction, respectively, in the antifungal activity against *B. cinerea* indicating that the N-terminal QSDKD and C-terminal INGF motifs are relatively minor contributors to the antifungal activity of this peptide. (Fig 6A).

## NCR13_PFV1 disrupts the plasma membrane of *B. cinerea* faster than NCR13_PFV2

Since NCR13_PFV1 exhibits ~17-fold more potent antifungal activity against *B. cinerea* than NCR13_PFV2, we hypothesized that the difference in antifungal activity lies in differential membrane permeabilization ability. To test this hypothesis, we performed SYTOX Green (SG) assays to examine the ability of NCR13_PFV1 and NCR13_PFV2 at 0.09 μM to permeabilize the plasma membrane of *B. cinerea* germlings. SG, a membrane-impermeant dye, binds to nucleic acids and is an indicator of membrane permeability. Average fluorescence intensities were measured along different time points and NCR13_PFV1 showed much higher

A

| Peptide | Sequence | MIC (µM) | Net Charge |
|---------|----------|----------|------------|
| NCR13_CS | TKPCQSDKDCKKFACRKPKVPKCINGFCKCVR | 3 | +8 |
| NCR13_AlaV1 | TKPC**AAAAA**CKKFACRKPKVPKCINGFCKCVR | 6 | +9 |
| NCR13_AlaV2 | TKPCQSDKDC**AAAA**CRKPKVPKCINGFCKCVR | Inactive (>12) | +6 |
| NCR13_AlaV3 | TKPCQSDKDCKKFAC**AAAA**VPKCINGFCKCVR | Inactive (>12) | +5 |
| NCR13_AlaV4 | TKPCQSDKDCKKFACRKPK**AAA**CINGFCKCVR | Inactive (>12) | +7 |
| NCR13_AlaV5 | TKPCQSDKDCKKFACRKPKVPKC**AAAA**CKCVR | 12 | +8 |

B

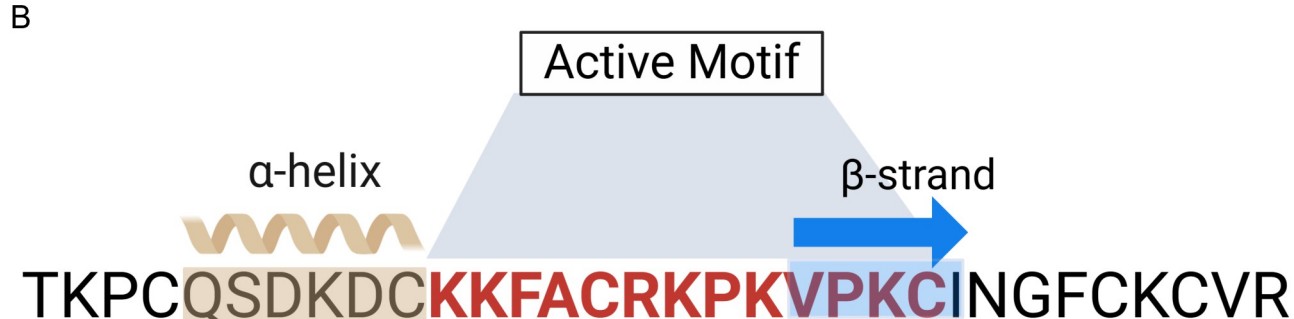

**Fig 6. Active motif of NCR13 required for antifungal activity.** (A) Antifungal activity of reduced NCR13_CS and reduced NCR13 alanine variants against *B. cinerea*. Alanine scanning mutagenesis was done by substituting alanines within the NCR13 core sequence, leading to the creation of the NCR13_AlaV1-V5. MIC values and net charge of all peptides were determined. The amino acid substitutions that completely inactivate the peptide are shown in red. Representative pictures are shown in the S6 Fig. (B) Active motif of NCR13 (highlighted with a bold red font) required for its antifungal activity against *B. cinerea*. The secondary structure elements are shown above the amino acid sequence. The brown wavy symbol and blue arrow indicate α-helix (α1) and β-strand (β1) respectively.

fluorescence intensity within exposed cells than NCR13_PFV2 (Fig 7A and 7B). After 10 min, SG uptake was observed in germlings challenged with NCR13_PFV1 but not in germlings challenged with NCR13_PFV2. After 20 min, a very strong SG uptake signal was observed in NCR13_PFV1 treated germlings, but only a modest signal in NCR13_PFV2 treated germlings (Fig 7A and 7B). Thus, NCR13_PFV1 inhibited fungal growth to a greater degree than NCR13_PFV2 at nanomolar concentration in part by permeabilizing the plasma membrane of *B. cinerea*.

## NCR13_PFV1 and NCR13_PFV2 reveal quantitative differences in their binding to multiple membrane phospholipids

Antifungal plant defensins gain entry into fungal cells and bind to a variety of bioactive phospholipids resident on the plasma membrane [9]. We speculated that NCR13_PFV1 and NCR13_PFV2 differ in their phospholipid binding specificity in the context of the membrane bilayer. Therefore, we used phospholipid strip assays to determine if NCR13_PFVs bind to different phospholipids. Our findings revealed that both NCR13_PFV1 and NCR13_PFV2

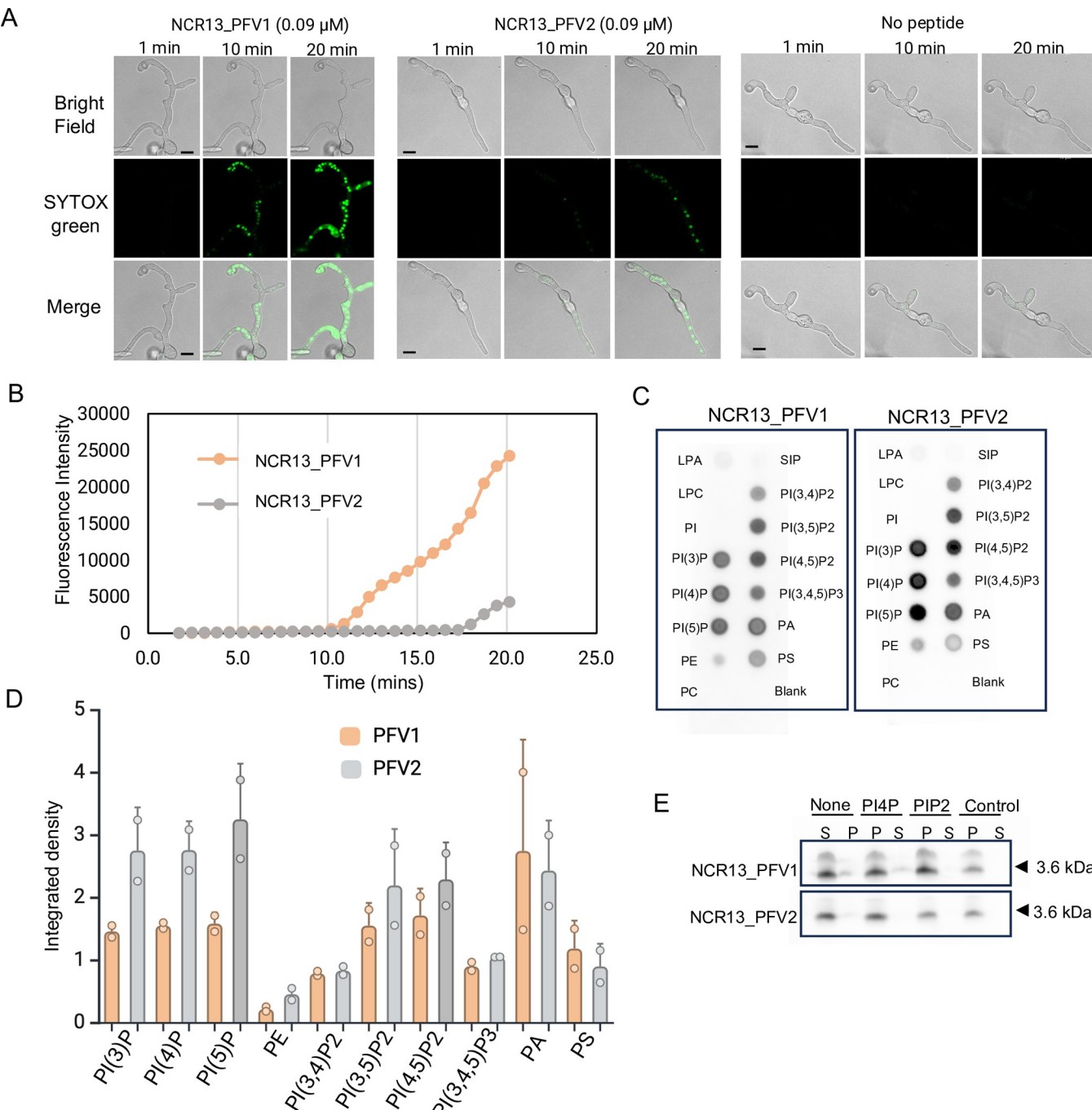

**Fig 7. NCR13 PFV1 permeabilizes the cell membrane of *B. cinerea* rapidly.** (A) Confocal microscopy images of SYTOX Green (SG) uptake in *B. cinerea* germlings at different time points. Germlings were treated with 0.09 μM NCR13_PFVs and SG dye simultaneously for 20 min. No peptide is used as a control. Scale bars = 10 μm. The experiment was repeated thrice with similar results. (B) Quantitative measurements of average fluorescence intensity over time to assess cell membrane permeability in *B. cinerea* germlings treated with 0.09 μM NCR13 PFVs. (C) PIP strip showing binding of NCR13 PFVs to multiple phospholipids, including phosphatidylinositol 4,5-bisphosphate (PI(4,5)P$_2$), phosphatidylinositol (3,4,5)-trisphosphate PI(3,4,5)P$_3$ and phosphatidylinositol monophosphates PI(3)P, PI(4)P, and PI(5)P, phosphatidylinositol bisphosphate PI(3,4)P$_2$. (D) Integrated density measurement of PIP strip probed with NCR13_PFVs. Datapoints are means ± SE of two independent biological replicates (*N* = 2). (E) Binding of NCR13_PFVs to PI(4)P and PI(4,5)P2 (PIP$_2$) containing liposomes. NCR13_PFV's were incubated without liposomes (None) and liposomes bearing no phosphoinositides (control), S- supernatant fraction and P- pellet fraction.

interacted with several phospholipids, including phosphatidylinositol-4-phosphate [PI(4)P] and PI(4,5)P$_2$ (Fig 7C). NCR13_PFV2 displayed significantly stronger binding affinity to phosphoinositides such as PI(3)P, PI(4)P, PI(5)P, PI(4,5)P$_2$, and PI(3,4,5)P$_3$ than NCR13_PFV1 (Fig 7D). The interaction of PI(4)P and PI(4,5)P$_2$ with NCR13_PFV1 and NCR13_PFV2 was further validated using a polyPIPosome binding assay (Fig 7E). Both NCR13_PFVs are positively charged and electrostatic interactions with the negatively charged plasma membrane are likely important for binding to membrane phospholipids. The modestly different distribution of these charges on the surface of these NCR13_PFVs (S7 Fig) may be responsible for the experimentally observed binding differences. Nevertheless, the significance of the phospholipid binding differences between the two peptides for their antifungal activity requires further investigation. Since several antifungal plant defensins bind to membrane phospholipids to oligomerize and induce membrane permeabilization [28], we tested the ability of NCR13_PFV1 and NCR13_PFV2 to oligomerize by incubating the peptides with the biochemical cross-linker bis(sulfosuccinimidyl)suberate (BS$_3$) in the presence and absence of PI(4,5)P$_2$. NCR13_PFV1 formed oligomers both in the presence and absence of PI(4,5)P2. On the other hand, NCR13_PFV2 formed higher order oligomers, but, only at higher concentrations (115 μM and 330 μM) of PI(4,5)P$_2$ (S8 Fig). Therefore, further investigation is required to determine whether phospholipid binding, including PI(4,5)P2, and the oligomerization of NCR13_PFVs are critical for antifungal activity.

## NCR13_PFV1 is internalized faster in germling cells of *B. cinerea* than NCR13_PFV2

Due to its much greater antifungal activity *in vitro* against *B. cinerea*, we hypothesized that NCR13_PFV1 exhibited faster entry into fungal cells at its MIC than NCR13_PFV2. To test this hypothesis, we exposed *B. cinerea* germlings to 0.09 μM DyLight550-labeled NCR13_PFV1 and NCR13_PFV2 and monitored their entry into germlings of *B. cinerea* using confocal microscopy. Within ~30 min, NCR13_PFV1 penetrated the germlings, however, NCR13_PFV2 did not enter the germlings (Fig 8A and 8B). At a concentration of 1.5 μM, NCR13_PFV1 started entering germlings in ~4 min, while NCR13_PFV2, after briefly gathering on the cell surface (Fig 8C, white arrowheads), started entering germlings at ~8 min. Unlike NCR13_PFV2's lack of entry at 0.09 μM, delayed entry of NCR13_PFV2 was evident at 1.5 μM (Fig 8C and 8D).

Following entry into *B. cinerea* germling cells, both NCR13_PFV1 and NCR13_PFV2 accumulated in distinct subcellular foci (Fig 8E, white arrows). Further localization studies using the nuclear staining dye Hoechst 33258 revealed that both peptides were localized in nuclei. To determine if each peptide was sub-localized within the nucleolus, we used the nucleolus-specific RNA staining dye Nucleolus Bright Green. As shown in Fig 8F, each peptide was localized within the nucleolus of germling cells suggesting that each peptide likely binds to ribosomal RNA (rRNA) or proteins and interferes with protein synthesis.

To gain further insight into the MoA of NCR13_PFV1 and NCR13_PFV2, we assessed the stability of each peptide once inside *B. cinerea* cells. We challenged *B. cinerea* germlings with each peptide separately for two different time points, 2 h and 4 h, and tested the stability of each peptide in fungal cells through western blot analysis. An NCR13 polyclonal antibody capable of binding to each peptide with equal affinity was used for this analysis (S9 Fig). NCR13_PFV1 accumulated in germling cells within 2 h and accumulated further after 4 h. In contrast, much less NCR13_PFV2 accumulated in germling cells within 2 h and was strongly decreased by 4 h (Fig 8G). Therefore, we highlight NCR13_PFV1 more rapidly enters fungal

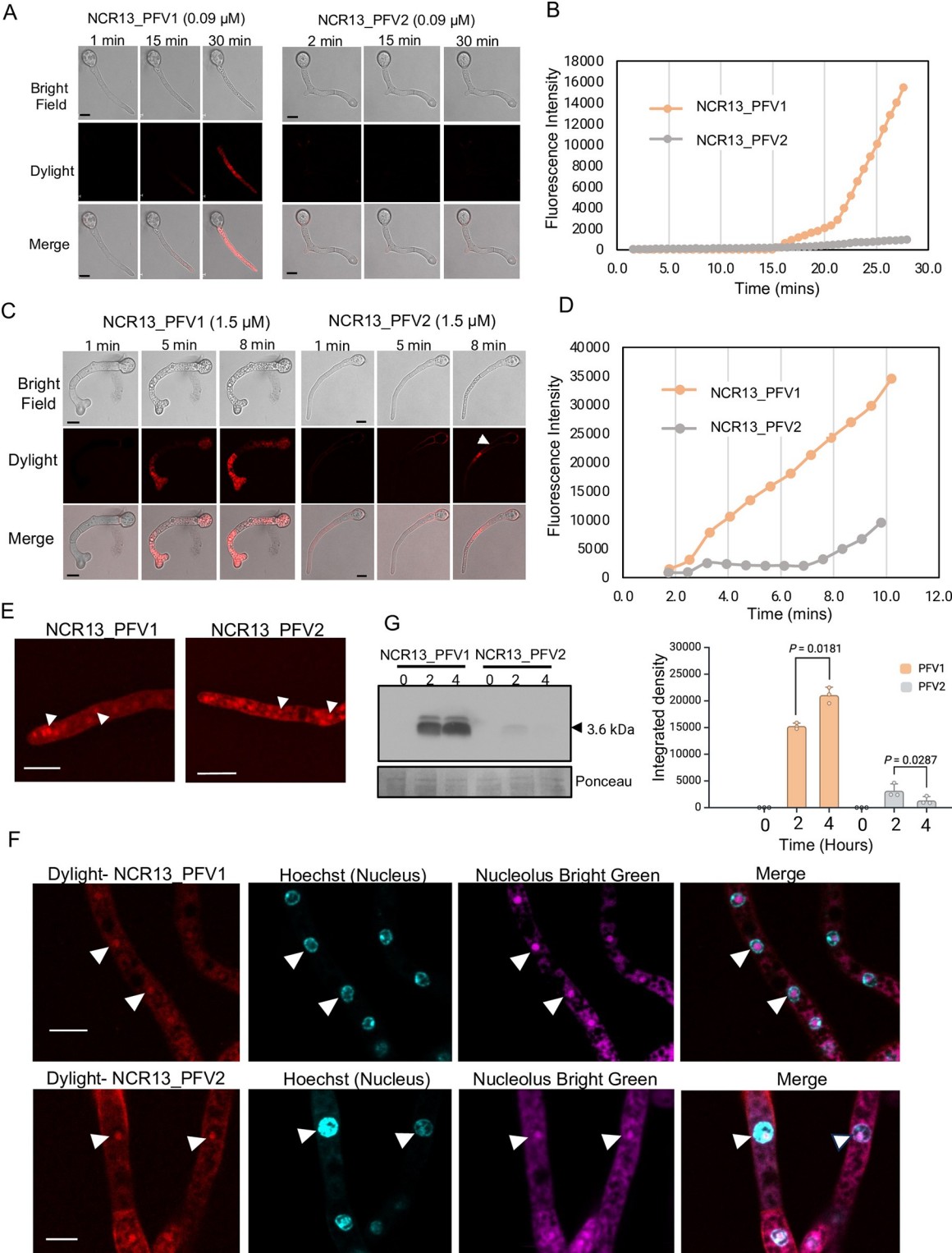

**Fig 8. Internalization and localization of NCR13 PFVs in *B. cinerea* germlings.** (A) Confocal microscopy images of *B. cinerea* germlings at different points treated with 0.09 μM DyLight550-labeled NCR13_PFV1 (left) and NCR_PFV2 (right) for 30 minutes. Scale bars = 10 μm. Data represents representative results of three independent experiments. (B) Quantification of peptide internalization was conducted by analyzing the average fluorescence intensities over time (in minutes) for NCR13_PFVs. The upper threshold for background was set at 23 for Fiji analysis. (C) Confocal microscopy images of *B. cinerea* germlings treated with 1.5 μM

DyLight550-labeled NCR13_PFVs for 30 minutes. White arrowheads indicate NCR13_PFV2 on the cell surface. Scale bars = 10 μm. (D) Quantitative measurements of peptide internalization in *B. cinerea* germlings treated with 1.5 μM NCR13_PFV1 (left) and NCR13_PFV2 (right). (E) Internalized DyLight550-labeled NCR13_PFV1 (left) and NCR13_PFV2 (right) concentrate in specific foci (white arrowheads) within *B. cinerea* germlings. Scale bars = 10 μm. (F) NCR13_PFV1 and NCR13_PFV2 localize to the nucleolus. DyLight550-labeled NCR13_PFV1 (upper panel) and PFV2 (lower panel) colocalize with rRNA-specific Nucleolus Bright Green stain. Hoechst 33258 is used for staining nucleus. White arrowheads indicate nucleolar localization. Scale bars = 5 μm. (G) Semi-quantitative analysis of NCR13_PFVs within *B. cinerea* germlings. Germlings were treated with 1.5 μM NCR13_PFV1 and NCR13_PFV2 for 0, 2, and 4 hours. Total proteins were extracted from fungal cells and immunoblotted using anti-NCR13 antibody. Equal loading of proteins was verified using ponceau staining. The bar graph represents integrated density measurements. Statistical significance was tested using paired Student's t-test. Data are shown as mean ± SD (N = 3, where N refers to biological replicates).

cells than NCR13_PFV2, and once inside NCR13_PFV1 is likely more stable than NCR13_PFV2.

## NCR13_PFV1 binds to rRNA and is a more potent inhibitor of protein translation *in vitro* than NCR13_PFV2

Since NCR13_PFVs are localized in the nucleoli of *B. cinerea*, we decided to test the binding of each peptide to rRNA. This was accomplished with an electrophoretic mobility shift assay (EMSA) [29] by incubating each peptide with total rRNA isolated from *B. cinerea* germlings. Our results showed that NCR13_PFV1 retarded the mobility of the rRNA at all concentrations greater than 0.18 μM, while NCR13_PFV2 showed no binding to rRNA until 3 μM (Figs 9A and S10). Therefore, it is evident that NCR13_PFV1 exhibits stronger binding affinity to fungal rRNA than NCR13_PFV2.

Next, we tested the ability of the NCR13_PFVs to inhibit protein translation *in vitro* because we hypothesized that NCR13_PFV1 may be a more potent inhibitor of protein translation than NCR13_PFV2. This was accomplished using a wheat germ *in vitro* translation system, which measured luciferase activity [30]. Protein translation inhibitory activity of each peptide was concentration dependent (Fig 9B). NCR13_PFV1 inhibited translation more effectively than NCR13_PFV2 at concentrations more than 0.75 μM. At 24 μM, NCR13_PFV1 almost completely halted translation compared to the ~50% inhibition by NCR13_PFV2 (Fig 9B). Based on these results, the difference in the antifungal potency of NCR13_PFVs may be partly due to the inhibition of protein translation *in vivo* with NCR13_PFV1 being a more potent protein translation inhibitor in *B. cinerea*.

## NCR13_PFV1 is more potent than NCR13_PFV2 semi-*in planta*

Next, we compared the semi-*in planta* antifungal activity of NCR13_PFV1 and NCR13_PFV2 using detached leaves of pepper as previously described [31]. To eliminate leaf-to-leaf variation, both peptides were added instantaneously with freshly prepared *B. cinerea* spores to the same leaf, along with no-peptide control, at different concentrations (3, 1.5, and 0.09 μM) (S11A Fig). Leaves were evaluated for reduction of gray mold lesions through measurement of photosynthetic efficiency Fv/Fm (variable fluorescence over saturation level of fluorescence) using CropReporter images. At 3 μM and 1.5 μM NCR13_PFVs, no disease symptoms were visualized (S11B Fig) and no significant difference in photosynthetic efficiency (S11C Fig) was observed, whereas the no-peptide control contained visible lesions and reduced photosynthetic efficiency. At 0.09 μM, only NCR13_PFV1 effectively suppressed the formation of visible lesions, while NCR13_PFV2 resulted in lesion sizes similar to the no-peptide control (S11B Fig). Moreover, a decrease in photosynthetic efficiency observed for NCR13_PFV1 was milder than observed for NCR13_PFV2. Together, this data further illustrates that NCR13_PFV1 is a more potent antifungal agent than NCR13_PFV2 in terms of preventing and reducing gray

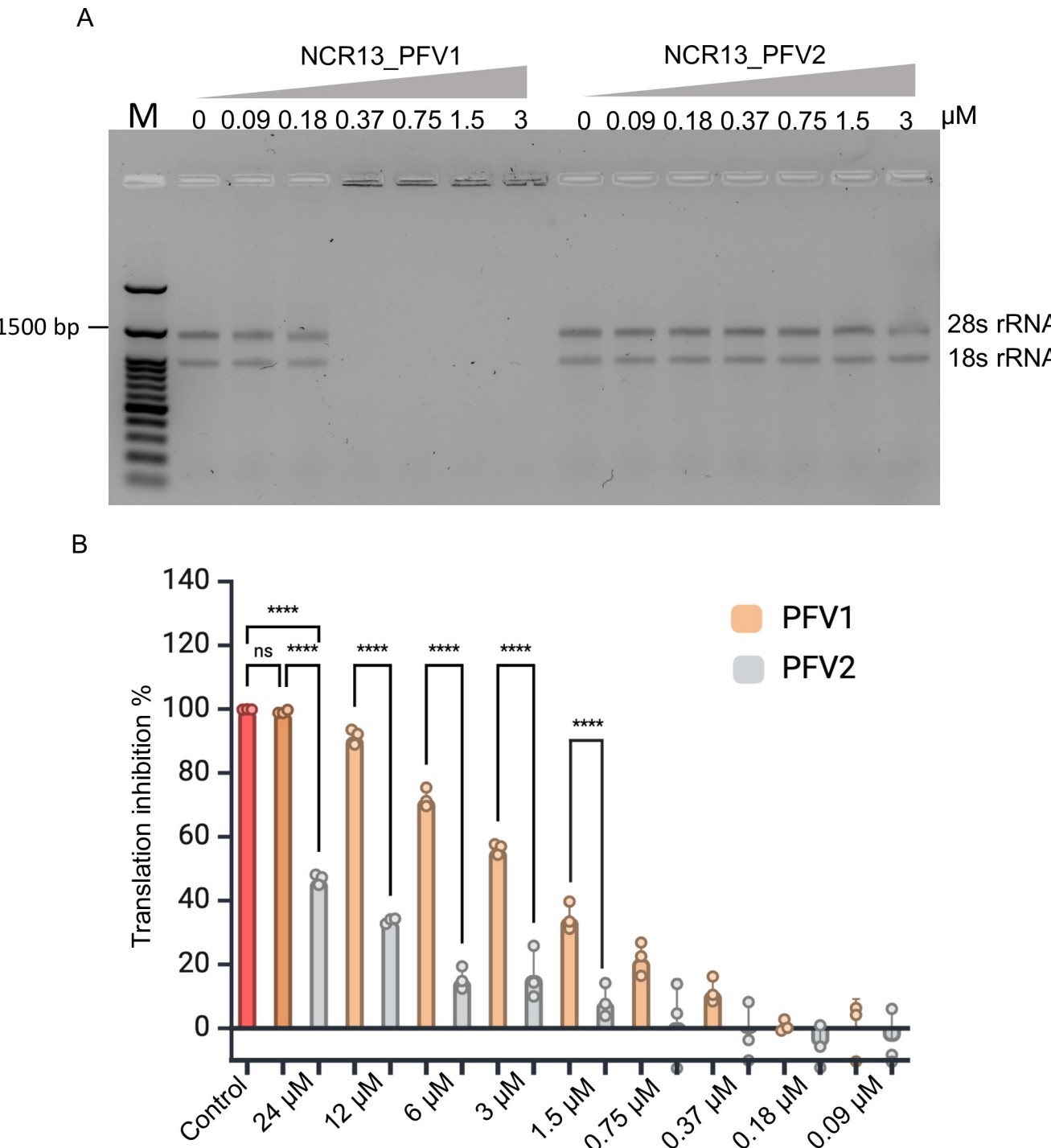

**Fig 9. NCR13_PFV1 exhibits rRNA binding and higher translation inhibition activity.** (A) rRNA binding by NCR13_PFVs tested using electrophoretic mobility shift assay (EMSA). *B.cinerea* rRNA was used to assess the binding activity of the peptides NCR13_PFV1 and NCR13_PFV2. The binding reactions were subjected to electrophoresis, and the mobility shifts were visualized on an agarose gel (1%). The concentration gradient of peptides is indicated above the lanes. The first lane on the left (M) contains molecular weight marker. (B) Relative *in vitro* translational inhibition (%) in the presence of NCR13_PFVs at different concentrations. Sterile water was used as negative control and cycloheximide (96 μM) was used as positive control. N = 3, 'N' indicates biological replicates. Statistical significance is determined by using One-way ANOVA with Tukey multiple comparisons test (**** indicates p value < 0.0001).

mold lesions on peppers (S11B and S11C Fig). These observations correlate well with the *in vitro* antifungal activity of each peptide. Based on this comparative study of their antifungal activity using a detached leaf assay, NCR13_PFV1 was judged to be a more promising peptide-based fungicide *in planta* (see below).

## Spray-applied NCR13_PFV1 is more effective in reducing gray mold disease symptoms in pepper and tomato plants

To determine if the *in vitro* and semi-*in planta* antifungal activity of these peptides translated to antifungal activity *in planta*, we tested the curative protection afforded by NCR13_PFV1 and NCR13_PFV2 against *B. cinerea* by spray application on pepper plants. Four-week-old pepper plants were pre-inoculated with freshly prepared *B. cinerea* spores. At 8 h post-inoculation, peptide at 1.5 and 3 μM concentration was sprayed onto pepper leaves. At 72 h post-inoculation, disease symptoms were evaluated by comparing photosynthetic efficiency (Fv/Fm) of the pathogen-challenged peptide-sprayed plants with the no-peptide control. Green areas indicate a high photosynthetic rate and healthy tissue, while yellow to red areas indicate a reduced photosynthetic rate and diseased tissue. As evidenced by their higher Fv/Fm values, leaves of pepper plants sprayed with NCR13_PFV1 at 1.5 and 3 μM had greater reduction in gray mold lesions than the leaves of pepper plants sprayed with NCR13_PFV2 at the same concentrations (Fig 10A and 10B). Pepper plants sprayed with 1.5 μM NCR13_PFV2 showed no significant protection (p>0.05) against gray mold when compared to the no-peptide control plants suggesting NCR13_PFV2 is unable to reduce gray mold disease symptoms at lower concentration (Fig 10A–10C). Correspondingly, the data aligns with the observation that NCR13_PFV1 markedly decreased the percentage of disease severity relative to NCR13_PFV2 at both treatment concentrations (Fig 10C). However, NCR13_PFV2 sprayed at 3 μM was able to provide significant curative protection compared to control plants (Fig 10A–10C), but it was not as effective as NCR13_PFV1. A similar observation was made when four-week-old tomato plants were also pre-inoculated with *B. cinerea* spores followed by spraying 3 μM of each peptide (S12A-S12C Fig). The NCR13_PFVs significantly decreased the disease severity, with NCR13_PFV1 demonstrating the most pronounced effect (S12B Fig). NCR13_PFV1 and NCR13_PFV2 inhibited gray mold disease 90% and 40%, respectively (S12C Fig). In summary, these data indicate the higher curative antifungal potential of NCR13_PFV1 for managing gray mold disease in pepper and tomato plants.

## Discussion

NCR13 is one of the highly cationic NCR peptides among the 63 found in chickpea nodules [10]. In this study, we discovered that NCR13 expressed in *P. pastoris* forms two isomers, NCR13_PFV1 and NCR13_PFV2, that share an identical amino acid sequence and molecular mass but differ in their disulfide pairing patterns. We report a comparative analysis of the structure, *in vitro*, *in planta*, and semi-*in planta* antifungal activity and MoA of NCR13_PFVs. Our results support the proposal that disulfide cross-linking provides an additional source of structural and functional diversity in NCR peptides [21].

Four- or six-cysteine NCRs are highly diverse in their primary sequences and are predicted to be structurally different. Previously, two solution NMR structures of *M. truncatula* NCR044 and NCR169, each containing two disulfide bonds, were solved [19, 23]. The NCR044 structure was largely disordered, dynamic, and contained a single α-helix and β-sheet [19]. Structural analysis revealed that the two NCR169 isomers contained only one short, two-stranded β-sheet, with one isomer being more extended and flexible while the other was more compact and constrained [23]. To date, a three-dimensional structure for a six-cysteine NCR peptide

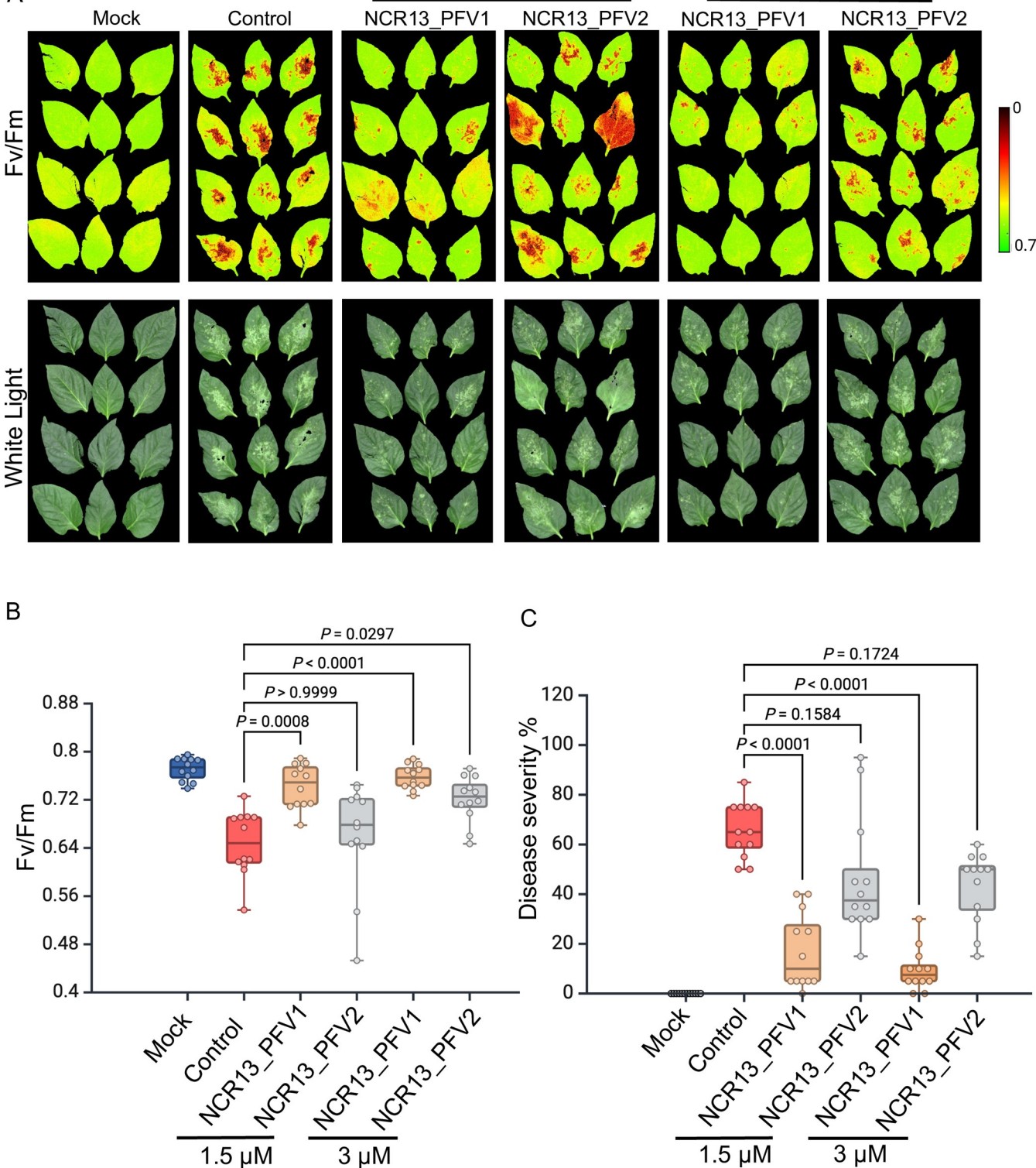

**Fig 10. NCR13 PFV1 exhibits enhanced curative antifungal activity against gray mold in pepper (*Capsicum annuum*) plants.** (A) Curative antifungal activity of NCR13_PFV1 and PFV2. Four-week-old pepper plants were first spray-inoculated using 1 mL of $5 \times 10^4$ *B. cinerea* spores. At 8 hpi (hours post inoculation), the plants were sprayed with 2 mL of peptide (NCR13_PFV1 or PFV2) at a concentration of 1.5 or 3 µM or control (*B. cinerea* alone). Mock-treated plants were only sprayed with SFM (no fungal spore suspension). (B) Calculated photosynthetic quantum yield (Fv/Fm) for individual pepper plants. (C) Disease severity (%) was calculated as described in materials and methods. (B-C) Each data point represents the average of 4 leaves per plant for 4 plants

for each treatment. Horizontal lines represent the median and boxes indicate the 25th and 75th percentiles. The Kruskal-Wallis test with Dunn's multiple comparisons test is used to determine statistically significant differences between control and peptide-treated inoculated plants. Two independent experiments were performed with similar results.

has not been reported. Here, we report the first solution structures of two NCR13 disulfide isomers each containing an α-helix and β-sheet. Nevertheless, unique pairing of two out of the three disulfide bonds significantly alters how these secondary structure elements fold upon each other, rendering NCR13_PFV1 more compact and resistant to digestion by trypsin or reduction by dithiothreitol. It is striking that *P. pastoris* generates two disulfide variants that are structurally different and raises the question of whether these variants are also expressed in the differentiating nodules of chickpea to perform different biological activities. In an earlier study, three disulfide variants of NCR247 containing four cysteines were generated using chemical synthesis and *in vitro* folding. NCR247 variants were found to have distinct patterns of biological activities such as bacterial gene expression, inhibition of bacterial cell division, protein translation, and antibacterial activity [21,22]. These observations suggest that NCR peptides can form disulfide variants *in vitro* or in an appropriate expression host, leading to diverse biological activities. Whether disulfide variants of NCR13, NCR247, or NCR169 are formed *in vivo* in nodules to control symbiosis in IRLC legumes remains an open question. In contrast to NCR peptides, plant defensins with eight cysteines expressed in *P. pastoris* are only observed as one species with an invariant disulfide pairing pattern [32].

To our knowledge, this study is the first detailed investigation of a chickpea NCR peptide and its antifungal action. Our data demonstrates that disulfide folding variants of NCR13 produced in *P. pastoris* exhibit profound differences in antifungal activity against multiple fungal pathogens of economic importance. NCR13_PFV1 possesses highly potent antifungal activity against *B. cinerea* and other pathogens, showing a four- to sixteen-fold greater activity compared to NCR13_PFV2. Remarkably, we found that the antifungal activity of NCR13_PFV1 is greater than other NCR peptides reported to date in the literature [33]. It is noteworthy that enhanced antifungal activity of both NCR13_PFVs produced in *P. pastoris* compared to chemically synthesized NCR13 without a disulfide bond underscores the critical role of disulfide bonds for antifungal activity. Furthermore, the formation of all three disulfide bonds is essential for highly potent antifungal activity of NCR13_PFV1 as removal of each disulfide bond resulted in a significant loss of its antifungal activity against *B. cinerea*. In addition, Cys residues are essential for the antifungal activity of NCR13 as their replacement with Ser leads to complete loss of antifungal activity. Acting as molecular anchors, disulfide bonds provide stability to the three-dimensional structure of proteins and peptides. They set the constraints for distance and angles between connected Cys residues, thus favoring the stability of the protein's folded state over its unfolded form [34]. Furthermore, the insertion of Cys residues into *de novo* designed peptides has produced hyperstable peptides with unique tertiary structures that form the basis for new peptide-based drugs [35,36]. The impact on antifungal activity of knocking out two instead of just one disulfide bond is yet to be investigated. Further, evaluating the three-dimensional structures of these double disulfide bond knockout variants will provide deeper insight into structure-activity relationships of this unique NCR peptide. *P. pastoris* has been the eukaryotic organism of choice for producing cysteine-rich peptides to ensure formation of the correct disulfide bond patterns [37]. It is intriguing that NCR13 expression results in the formation of two disulfide folding variants in *P. pastoris*. In this context, it is critical to observe that knocking out either the C4-C23 or C15-C30 disulfide bond in NCR13_PFV1 results in the formation of one peak on HPLC chromatograms, whereas knocking out the C10-C28 bond still results in the formation of two distinct peaks. Mass

spectrometry analysis shows that peptides from two HPLC peaks of the C10-C28 knockout and one HPLC peak in the C4-C23 and C15-C30 knockouts are in the oxidized state. Therefore, it is likely they are in a folded/oxidized form. Finally, understanding the molecular mechanisms governing the generation of disulfide variants of specific antifungal NCR peptides could pave the way for production of targeted disulfide structural variants in *P. pastoris* with enhanced antifungal activity.

Antifungal peptides with potent fungicidal activity and multiple MoA are attractive candidates for development as spray-on fungicides. NCR13 has a promise as one such candidate. For instance, the ability to permeabilize the plasma membrane of fungal pathogens is a common MoA of plant cysteine-rich antifungal peptides including NCRs [14]. NCR13_PFV1 can permeabilize the plasma membrane of *B. cinerea* rapidly at very low concentrations. The membrane disrupting ability of NCR13_PFV1 is far greater than that of NCR13_PFV2 and NCR044 [19]. Both NCR_PFVs bind to the same phospholipids on lipid strips with different binding intensity that may be related to the modest difference in the distribution of charges on the peptides' surface. However, it is not clear what role, if any, phospholipid binding plays in conferring increased membrane disrupting ability of NCR13_PFV1 and this aspect remains to be elucidated. In addition, NCR13_PFV1 gains rapid entry into the interior of fungal cells compared to NCR13_PFV2. It is plausible that more compact folding of NCR13_PFV1 allows it to more quickly penetrate the cell wall and pass through the outer and inner leaflets of the plasma membrane. However, the possibility remains that these two peptides use different mechanisms to cross the fungal cell wall and disrupt the plasma membrane. Further studies are needed to determine if these two peptides use different energy-dependent or energy-independent, and concentration-dependent pathways for entry into fungal cells. It also remains to be determined if the entry of each peptide into fungal cells requires endocytosis. Notably, *B. cinerea* germlings challenged with 1.5 μM of each peptide showed NCR13_PFV1 enters fungal cells in much greater levels after 2 h than NCR13_PFV2. Moreover, once inside the cells, NCR13_PFV1 accumulates stably for at least 4 h whereas NCR13_PFV2 tends to degrade in this time. Greater stability of NCR13_PFV1 to proteolysis might be a major contributing factor to its much greater antifungal activity against this pathogen, an observation that warrants further investigation.

Subcellular localization studies showed that NCR_PFVs are targeted to nucleoli in the germlings of *B. cinerea* suggesting that these peptides are targeting ribosomal biogenesis or protein translation machinery as part of their MoA as proposed earlier for the MoA of NCR044 [19]. In support of this hypothesis, we found that addition of each NCR13_PFVs to the *in vitro* translation system inhibited protein synthesis in a concentration dependent manner. NCR13_PFV1 proved to be more effective in inhibiting protein translation *in vitro*. NCR13_PFV1 also binds to *B. cinerea* rRNA at a lower concentration (0.37 μM) than NCR13_PFV2 (6 μM). Further research is needed to determine if differences in rRNA binding affinity and protein translation inhibition *in vitro* directly translate to profound differences in their antifungal activity *in vivo*. NCR peptides appear to target protein translation as part of their antimicrobial mechanism as evidenced by *M. truncatula* NCR247 inhibiting protein translation *in vitro* by binding to ribosomal proteins [21,38]. Plant defensin MtDef4 also inhibits protein translation *in vitro* by binding to ribosomal proteins, but not to ribosomal RNA [39]. Besides that, it is likely that NCR13_PFV1 and NCR13_PFV2 have different, yet unidentified, intracellular targets which will be the subject of future investigation.

Fungicides currently used commercially for the management of fungal diseases target ergosterol biosynthesis, succinate dehydrogenase, and the cytochrome $bc_1$ complex. These targets are prone to rapid development of fungal resistance [40]. Small NCR peptides with multifaceted MoA have emerged as potential candidates for development as bioinspired fungicides.

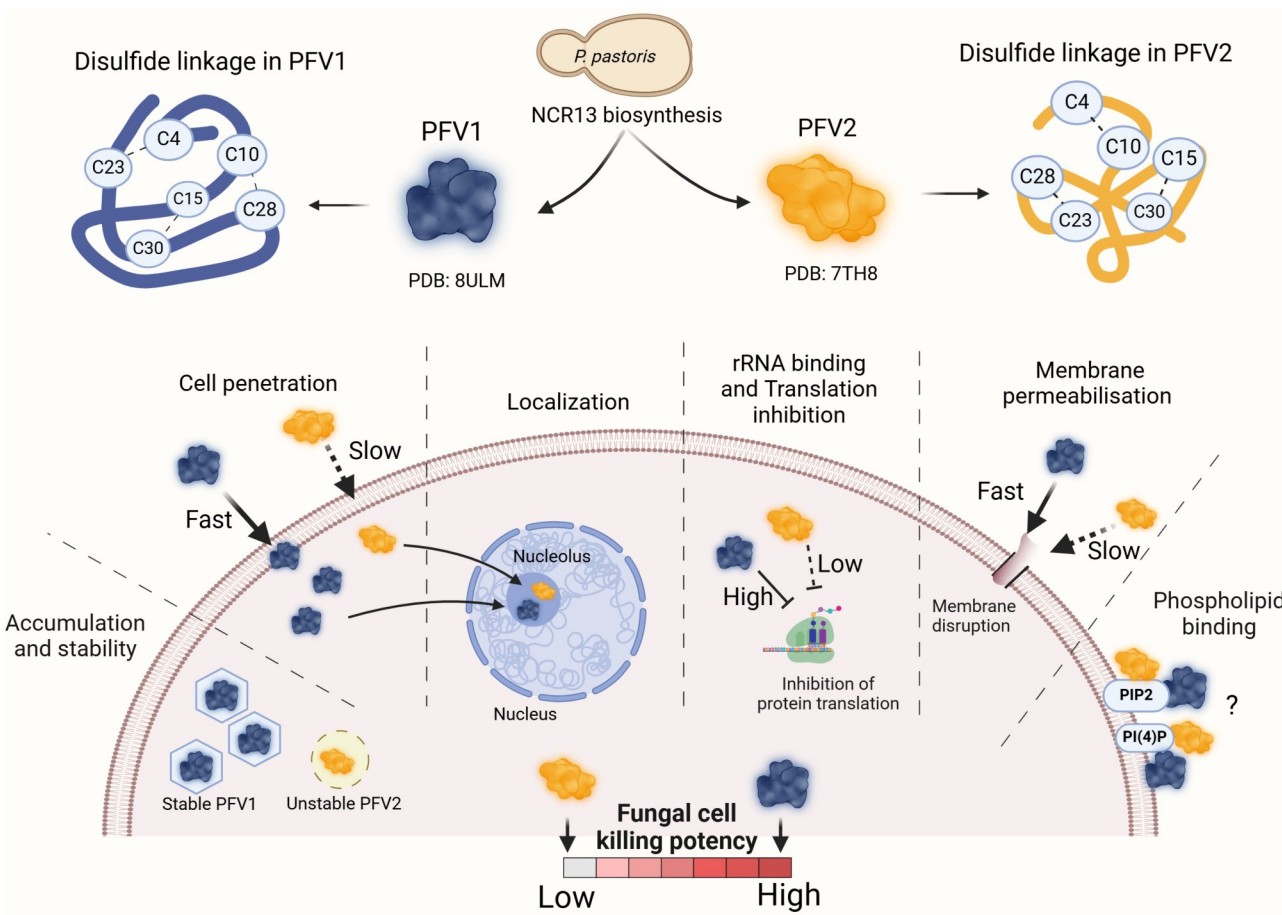

**Fig 11. Graphical abstract illustrates the biosynthesis of disulfide linkage variants of NCR13 (PFV1 and PFV2) in *P. pastoris*, their structures, antifungal activity, and modes of action in *B. cinerea* germlings.** The figure depicts the differences in cell penetration, localization, rRNA binding and translation inhibition, membrane permeabilization, phospholipid binding, and overall fungal cell killing potency. Created using BioRender.com.

We have previously demonstrated efficacy of *M. truncatula* NCR044 as a spray-on fungicide for management of gray mold in tomato [19]. Here, we report NCR13 is also a multi-target antifungal peptide that reduces gray mold disease symptoms curatively when sprayed on pepper and tomato plants. Consistent with its *in vitro* and semi-*in planta* antifungal activity, NCR13_PFV1 affords greater protection against gray mold disease than NCR13_PFV2 in both pepper and tomato plants. NCR13 is amenable to large scale recombinant production in a bio-fermenter using a *P. pastoris* expression system. However, further characterization of NCR13 is needed, including demonstrating its efficacy for control of gray mold and other diseases in large-scale greenhouse and field tests, assessment of off-target activity against mammalian cells, and phytotoxicity. In summary, this study demonstrated that disulfide bond pairing pattern has a profound impact on the *in vitro* antifungal activity, MoA, and *in planta* antifungal activity of chickpea NCR13 (Fig 11). Overall, our work underscores the significance of disulfide cross-linking in enhancing the efficacy of antifungal peptides for agricultural use.

## Materials and methods

### Fungal cultures and spore suspension

*B. cinerea* strain T4 was cultured in V8 medium for 3 weeks at 25°C. Fungal spores were collected by flooding the fungal growth plates with water and scraped using a sterile spreader.

The spore suspension was filtered through two layers of Miracloth, followed by centrifugation at 12,000 g for 2 minutes (min) and subsequently washed twice with sterile water. Lastly, fungal spores were resuspended in Synthetic Fungal Media (SFM) [41]. The spore suspension was adjusted to the desired spore density of about $10^5$ spores/mL using a hemocytometer. *Fusarium virguliforme (F. virguliforme)* was grown and maintained regularly in potato dextrose agar (PDA) (Difco Laboratories) medium. Two weeks old *F. virguliforme* plates were used to collect spores as described above for *B. cinerea* followed by centrifugation and washing steps. Spores were counted using a hemocytometer and adjusted to the final concentration of $10^5$ spores/mL in 2X SFM media. *Sclerotinia sclerotiorum* obtained from the field of pennycress in Illinois was grown on PDA medium at room temperature.

## Expression and purification of NCR13

For expression in *P. pastoris*, a codon-optimized gene (synthesized by GenScript) was created based on the published amino acid sequence of the mature NCR13 peptide, including an additional alanine at the N-terminal derived from the signal peptide [24] The NCR13 gene was expressed in *P. pastoris* X33 using either a SacI linearized pPICZα-A integration vector (Invitrogen) or at XhoI and XbaI restriction sites. NCR13 was purified using a CM Sephadex C-25 cation-exchange chromatography as described [19]. After induction, cell-free supernatant was collected after centrifugation (10,000×g) at 4˚C for 20 min, and the pH was adjusted to 6.0. The supernatant was passed through cation-exchange resin, CM Sephadex C-25 (Sigma-Aldrich, Cat No: C25120) previously equilibrated with binding buffer (25 mM sodium acetate anhydrous, pH 6.0) using an AKTA FPLC. Peptides bound to the resin were eluted using the elution buffer (1M NaCl, 50 mM Tris, pH 7.6). The FPLC fraction containing NCR13 was concentrated and desalted using an Amicon stirred cell with a 3-kD cutoff membrane (Millipore, Cat No: PLBC04310) and further purified by reverse phase C18-HPLC. HPLC fractions collected separately for NCR13_PFV1 and NCR13_PFV2 were lyophilized and re-suspended in nuclease-free water. Peptide concentrations were determined using the Pierce BCA Protein Assay Kit (Thermo Scientific). NCR13 and alanine variants of NCR13 were chemically synthesized (Biomatik) and further purified by in-house HPLC.

## Molecular weight determination

Purified peptides were analyzed by LC-MS for accurate mass determination. Peptide samples were cleaned using Thermo Scientific HyperSep C18 microscale SPE extraction tips and, after dry-down, resuspended in 0.1% formic acid for LC-MS analysis. LC-MS was performed with a Dionex RSLCnano HPLC coupled to an OrbiTrap Fusion Lumos (Thermo Scientific) mass spectrometer using 80 min gradient (1–90% acetonitrile). Sample was resolved using a 75 μm x 15 cm PepMap C18 column (Thermo Scientific). MS spectra of protein ions of different charge-states were acquired in positive ion mode with a resolution setting of 120,000 (at *m/z* 200). Mass deconvolution was performed using ProteinProspector, Xcalibur (Thermo Scientific), and ESIprot Online.

## Disulfide bond pattern identification for NCR13_PFV2 using mass spectrometry

An aliquot of unlabeled, oxidized NCR13_PFV2 was mixed with 20-fold less (in mass) trypsin, incubated at room temperature for 1 h, and diluted in 50% acetonitrile, 0.1% formic acid to ~0.1 μM. The diluted samples were infused at 2 μL/min into a Thermo Q-Exactive Exploris 480 Orbitrap mass spectrometer with positive mode electrospray at 3.4 kV. The ion transfer tube was at 320˚C, sheath gas at 2 psi, and auxiliary gas at 5 psi. Data-dependent acquisition

was used (MS1 resolution 60k, MS2 resolution 30k) over 8 min. Fragmentation spectra were collected with higher-energy collisional dissociation (HCD) at 30% collision energy. Data were analyzed and deconvoluted in FreeStyle (Thermo Scientific). Crosslinked peptides were identified using Byonic (ProteinMetrics). Under the same conditions, NCR13_PFV1 could not be digested by trypsin.

## $^{15}$N isotopic labeling of NCR13 for NMR structural analysis

The $^{15}$N-labeled NCR13_PFVs were prepared according to the protocol as described previously [32] with minor modifications. NCR13-expressing *P. pastoris* X33 was grown at 28 ± 2˚C for two days on a YPD plate containing 150 µg/mL Zeocin. A single colony was then inoculated into 5 mL of YPD broth and grown overnight at 28 ± 2˚C on a rotary shaker at 225 ± 25 rpm. After overnight growth, a 5 mL culture was transferred aseptically to 100 mL of FM23 medium (0.8% YNB without amino acids and ammonium sulfate, 2 mg/L biotin, 1.2% $(NH_4)_2SO_4$, 0.3% $K_2HPO_4$, 0.28% $KH_2PO_4$, 2 ml/L PTM1 salts) containing 3% glucose. After 24 h, an additional 2% glucose was added to the culture medium. After a growth phase of ~30 h and just prior to methanol induction, the culture medium was adjusted to 0.02% $^{15}$N-ammonium chloride ($^{15}NH_4Cl$) and 0.1% glucose. After 40 h, cells were harvested by centrifugation at 5,000 rpm at room temperature for 10 min, briefly washed with 0.2% glycerol, pelleted, and re-suspended in 100 mL FM23 medium containing 1.2% $^{15}NH_4Cl$ and grown for 4–5 days at 25˚C at 225 ± 25 rpm. In the induction phase, a low concentration of 0.2% methanol was initially added to the culture medium to induce protein expression and gradually increased to 0.4% from 72 to 120 h (1.4% final). The labeled NCR13 peaks were purified from the *P. pastoris* growth medium following the same protocol used for the unlabeled peptide.

## NMR spectroscopy of NCR13

All NMR data were collected at 20˚C on $^{15}$N-labeled samples of oxidized NCR13_PFV1 and NCR13_PFV2 (~ 1 mM, 20 mM sodium acetate, 50 mM NaCl, pH 5.3) using Varian (Inova-600) or Bruker (Advent-750) spectrometers equipped with an HCN-probe and pulse field gradients. Except for the carbonyl resonances (which could not be assigned because making $^{13}$C-labeled samples in *P. pastoris* was too expensive), the $^1$H, $^{13}$C, and $^{15}$N chemical shifts of the backbone and side chain resonances were assigned from the analysis of two-dimensional $^1$H-$^{15}$N HSQC, $^1$H-$^{13}$C HSQC, $^1$H-$^1$H TOCSY, $^1$H-$^1$H NOESY, and $^1$H-$^1$H COSY spectra and three-dimensional $^{15}$N-edited NOESY-HSQC (mixing time = 90 ms) and $^{15}$N-edited TOCSY spectra. To obtain additional chemical shift assignments, inter- and intra-residue proton side chain restraints for protons near the residual water resonance, most of the two-dimensional data sets were also collected by lyophilizing and resuspending the sample in 99.8% $D_2O$. In preparing the latter sample, a $^1$H-$^{15}$N HSQC spectrum was collected ~10 min after the addition of $D_2O$ to identify slowly exchanging amides that could be used as hydrogen bond restraints in the structure calculations. To confirm oxidized NCR13_PFV1 and NCR13_PFV2 arose from different disulfide bond patterns of a common precursor primary amino acid sequence, both oxidized samples were reduced by making the NMR samples 5 mM in dithiothreitol (DTT). While NCR13_PFV2 was reduced rapidly, it took more than 5–7 days to fully reduce NCR13_PFV1. All NMR data were processed using Felix 2007 (MSI, Molecular Simulations) software and analyzed with the program Poky (v3.115) [42]. The assigned $^1$H, $^{13}$C, and $^{15}$N chemical shifts were referenced using indirect methods (DSS = 0 ppm) and deposited into the BioMagResBank database (www.bmrb.wisc.edu) under the accession numbers:31111 (oxidized NCR13_PFV1 8ULM) and 30979 (oxidized NCR13_PFV2 7TH8).

## NMR solution structure calculations for oxidized NCR13_PFV1 and NCR13_PFV2

The protocols used to obtain solution structures for both peptides followed a similar approach with the $^1$H and $^{15}$N chemical shift assignments and peak-picked NOESY data serving as initial experimental inputs in iterative structure calculations with the program CYANA (v 2.1) [43]. The $^1$H, $^{13}$C, and $^{15}$N chemical shift assignments were also the primary basis for the introduction of 24 (NCR13_PFV1) and 25 (NCR13_PFV2) dihedral Psi (ψ) and Phi (φ) angle restraints identified with the program TALOS+ using the online webserver (https://spin.niddk.nih.gov/bax/nmrserver/talos/) [44]. Nine restraints between the side chain sulfur atoms of C4-C23, C15-C30, and C10-C28 (NCR13_PFV1) and C4-C10, C15-C30, and C23-C28 (NCR13_PFV1) (2.0–2.1 Å, 3.0–3.1 Å, and 3.0–3.1 Å for the Sγ-Sγ, Sγ-Cβ, and Cβ-Sγ distances, respectively) [19, 45] were introduced on the basis of AlphaFold predictions for NCR13_PFV1 and high resolution mass spectrometry analysis for NCR13_PFV2. Exhaustive attempts to identify the disulfide bond pattern for NCR13_PFV1 using various mass spectrometric methods failed likely due to its hyperstability to reduction by DTT or tris(2-carboxyethyl) phosphine. Backbone hydrogen bond restraints (1.8–2.0 Å and 2.7–3.0 Å for the NH–O and N-O distances, respectively) were added to the calculations for proximate backbone amide-carbonyl groups present in early structure calculations that were accompanied with slowly exchanging amides in the deuterium exchange experiment. The final ensemble of 20 CYANA derived structures was refined in explicit water with CNS (version 1.1) using the PARAM19 force field and force constants of 500, 500, and 1000 kcal for the NOE, hydrogen bond, and dihedral restraints, respectively. For these water refinement calculations, 10 and 0% were added to the upper boundary of the CYANA distance restraints for NCR13_PFV1 and NCR13_PFV2, respectively, and the lower boundary set to the van der Waals limit for both. Structural quality was assessed using the online Protein Structure Validation Suite (PSVS, v1.5) [46]. Complete structural statistics are summarized in S3 Table. The atomic coordinates for the final ensemble of 20 structures have been deposited in the Research Collaboratory for Structural Bioinformatics (RCSB) under PDB codes 8ULM (NCR13_ PFV1) and 7TH8 (NCR13_ PFV2).

## Peptide uptake and stability assessment within *B. cinerea*

*B. cinerea* (~$10^5$ spores/mL) was inoculated in 10 mL of Potato Dextrose Broth (PDB) and incubated in a shaker at 28°C for 18 h. *B. cinerea* germlings were treated with 1.5 μM NCR13_PFV1 and NCR13_PFV2 for 0, 2, and 4 h. Total proteins were extracted from the fungal cells using 300 μL of HEPES buffer (100 mM HEPES, 5mM EDTA, 0.1% Triton-100, Pro-Block Gold Yeast/Fungi Protease Inhibitor, pH 7.5) and centrifuged at 14000 rpm to remove cell debris. The concentration of protein present in the supernatant was measured using Pierce BCA Protein Assay Kit (Thermo Scientific). Total proteins (10 μg) were separated on a 4–20% SDS-PAGE gel (Bio-Rad, TGX Stain-Free precast gels) and transferred to a nitrocellulose membrane. Equal loading of proteins was checked using Ponceau S staining (Thermo Scientific). Immunoblotting was performed using the custom synthesized anti-NCR13 polyclonal antibody (Biomatik) at a concentration of 0.1 μg/mL followed by an incubation with the secondary anti-rabbit antibody (Cytiva, Cat No: RPN4301) at a dilution of 1:20,000.

## *In vitro* antifungal assay

The antifungal activity of NCR13 against fungal pathogens was determined using the resazurin cell viability assay, as described previously [47]. Briefly, 45 μL of each peptide dilution (0, 0.09, 0.1875, 0.375, 0.75, 1.5, 3, and 6 μM) was added to each well of the microtiter plate containing

45 μL (~$10^5$ spores/mL) of spore suspension. Three biological replicates were used for each treatment and the experiments were repeated three times. Resazurin at 0.1% (w/v) was added to each well after incubation for 48 h. Images were taken at ~24 h. The Minimum Inhibitory Concentration (MIC) for each peptide denotes the lowest concentration at which the peptide halts the growth of fungus evident through a lack of color change from blue to colorless. The *in vitro* antifungal activity of NCR13_PFVs against *S. sclerotiorum* was determined using a 24-well plate assay as previously described [48]. Mycelial plugs (1 mm in diameter) cut from the growing edge of the two day-old actively growing colony were transferred from PDA media to a series of wells containing 250 μL of SFM.

## Internalization and permeabilization assays using confocal microscopy

NCR13_PFV1 and PFV2 were labeled with DyLight550 amine reactive fluorescent dye following the manufacturer's instructions (Thermo Scientific). *B. cinerea* germlings (50 μL, ~$5 \times 10^4$ spores/mL) were treated with DyLight 550 labeled NCR13_PFV1 and NCR13_PFV2 (50 μL, 0.09 μM and 1.5 μM), and the fluorescence measured with excitation and emission wavelengths of 550 and 560–600 nm, respectively. Time-lapse confocal microscopy data with a z-stack at each time point were acquired using a 63X water immersion objective lens (HC PL Apochromat CS2) at 30-second intervals. For the membrane permeabilization assay, fluorescent dye SYTOX Green (Thermo Scientific) at a concentration of 0.25 μM was used to determine the membrane permeabilization of *B. cinerea* germlings as described previously [31]. Time lapse imaging was performed to visualize the SYTOX Green entry upon treatment with 50 μL of 0.09 and 1.5 μM NCR13_PFVs. Imaging was performed on a Leica SP8 confocal microscope with excitation at 488 nm and emission at 508–544 nm. Average fluorescence intensity measurements were obtained using Fiji [49].

## Phospholipid binding assays

Phospholipid binding of NCR13_PFVs was performed with PIP Strips (Echelon Biosciences, Cat No: P-6001) that were spotted with 100 pmol of various biologically active phospholipids. PIP strips were incubated with 1 μg/mL NCR13_PFVs at 4°C for 60 min. Peptide binding to the membrane strips was detected using monoclonal NCR13 antibody (0.1 μg/mL) followed by goat anti-rabbit IgG HRP (Cytiva RPN4301) at 1:20,000 dilution. The binding signals were detected using the SuperSignal West Pico Chemiluminescent Substrate Kit (Thermo Scientific) following the manufacturer's protocol. Quantitative analysis of the blot intensity was performed using Fiji [49].

## Oligomerization assay

To determine the oligomerization status of the NCR13_PFVs, the peptides were prepared in PBS buffer at 1.5 mg/mL (5 μL/reaction) and incubated with specified concentrations of PI [4,5]P2. The cross-linking reaction was initiated by addition of 12.5 mM bis(sulfosuccinimidyl)suberate BS3 (Thermo Scientific) and then incubated for 30 min at room temperature. After incubation, tricine sample buffer was added to each sample and heat denatured at 95°C for 5 min. Samples were loaded on a 16.5% Tris-Tricine precast gel (Bio-Rad Laboratories). For SDS-PAGE, 5 μL of Precision Plus Dual Xtra Protein Standards (2–250 kDa) were used (Bio-Rad Laboratories). Oligomerization patterns were visualized using silver staining.

## PolyPIPosome binding assay

The validation of the binding of NCR13_PFV1 and NCR13_PFV2 to phospholipids was further performed using a liposome binding assay. In this assay, polyPIPosome binding was

conducted in 200 μL of binding buffer (20 mM HEPES pH 7.4, 120 mM NaCl, 1 mM EGTA, 1 mM $MgCl_2$, 1 mg/mL BSA, 0.2 mM $CaCl_2$, 5 mM KCl) plus 10 μL each of control PolyPIPosome, PI4P PolyPIPosomes, and $PIP_2$ PolyPIPosome (Echelon Biosciences). Subsequently, 2 μg of each peptide was added, and the mixture incubated for 1 h at room temperature. Following incubation, liposomes were collected by centrifugation at $16,000 \times g$ for 20 min, and the supernatant was collected. The pellet was subjected to three washes in 1 mL of binding buffer. After the washes, the pellet was resuspended in 40 μL of 2X Laemmli sample buffer. Both samples were reduced with 100 mM DTT and separated on a 4–20% TGX Stain-Free precast gels (Bio-Rad Laboratories). Proteins were then transferred to a nitrocellulose membrane and subjected to blotting using the NCR13 antibody.

### Electrophoretic mobility shift assay

Total RNA was extracted and purified from *B. cinerea* grown in Potato Dextrose Broth (Fisher Scientific) using the NucleoSpin RNA Plus Mini kit (Takara Bio) according to manufacturer instructions. The gel shift experiments were performed by mixing 200 ng of the total RNA with different concentrations of NCR13 PFV1 and PFV2 (0–3 μM) in 20 μL of binding buffer (10 mM Tris-HCl, pH 8.0, 1 mM EDTA). The reaction mixtures were incubated for 1 h at room temperature and mixed with 4 μL of 6X gel loading dye (New England Biolabs). Following electrophoresis in a 1.2% agarose gel containing SYBR safe DNA gel stain (Thermo Scientific) with TAE buffer at 120 V for 45 min, the gels were visualized using a Bio-Rad ChemiDoc XRS+ system.

### *In vitro* translation assay

*In vitro* translation was performed using the RiboMAX Large Scale RNA Production system and Wheat Germ Extract *in vitro* translation system (Promega). The luciferase DNA template (Promega) was transcribed to mRNA and translated to luciferase following the manufacturer's protocol. Luciferase enzyme activity was detected with the luciferase kit (Promega) using a luminometer (Tecan). Different concentrations of NCR13_PFV1 and NCR13_PFV2 were used to determine their translation inhibition ability. Sterile water was added to the *in vitro* translation system as a negative control and a fungal translation inhibitor cycloheximide (48 μM) was used as a positive control. Translation inhibition (TI) percentage was calculated using this formula TI % = $(1- a/b) \times 100$, where 'a' equals the translation level with peptide and 'b' equals the translation level without peptide.

### *In planta* and Semi-*In planta* antifungal assays

Pepper (*Capsicum annuum*, California Wonder 300 TMR) plants used in this study were grown in a greenhouse for 4 weeks under 14 h light/10 h dark cycles. Tomato (*Solanum lycopersicum* cv. Mountain Spring) plants were grown in a controlled environment growth chamber for 4 weeks, under 16 h light/ 8 h dark cycles. Semi-*in planta* antifungal activity of each peptide against *B. cinerea* was determined using detached leaves of pepper. NCR13_PFV1 and NCR13_PFV2 were tested at 6, 1.5, and 0.09 μM. Following incubation of 10 μL peptide with 10 μL of *B. cinerea* spores (~$10^5$ spores/mL) under a humid environment for 48 h, the leaves were photographed in white light. High-resolution fluorescence images were also taken using CropReporter (PhenoVation). These images depicted the calculated Fv/Fm (maximum quantum yield of photosystem II) values of the diseased area affected by *B. cinerea* infection. Colors in the images show five different classes ranging from class I to class V (0.000 to 0.700) depicting varying degrees of tissue damage.

 

To test the curative antifungal activity of peptides against *B. cinerea*, 4-week-old pepper and tomato plants were used. Plants were first sprayed with 1 mL of *B. cinerea* spores (about $5 \times 10^4$) suspended in SFM. After 8 h of *B. cinerea* challenge, 2 mL of NCR13_PFV1 or NCR13_PFV2 (3.0 and 1.5 μM) were sprayed per plant. Control plants were sprayed with 2 mL of water. All plants were placed in a high humid environment for 72 h. Plants were imaged at 72 h after infection in white light and high-resolution fluorescence images were taken using CropReporter (PhenoVation). The quantification of plant health was carried out using the calculated Fv/Fm (maximum quantum yield of photosystem II) images of CropReporter showing the efficiency of photosynthesis in false colors.

For the curative assay using pepper and tomato plants, pictures of whole plants and detached leaves were taken as described above. The disease symptoms on individual leaves of each plant were visually assessed in % of disease portion (disease severity) and on a rating scale of 0–5 where each rank would represent the following proportion of diseased tissue per leaf: Scale 0: No obvious signs/symptoms of disease, Scale 1: $0 > x \leq 20\%$ diseased portion of leaf, Scale 2: $20 > x \leq 40\%$ diseased portion of leaf, Scale 3: $40 > x \leq 60\%$ diseased portion of leaf, Scale 4: $60 > x \leq 80\%$ diseased portion of leaf, Scale 5: $80 > x \leq 100\%$ diseased portion of leaf. Four plants were used per treatment and for each plant the disease was assessed per leaf. Besides disease severity (DS), the inhibition of disease by treatment was also evaluated. The following formula was used to assess the inhibition of disease severity by each treatment: % Disease Inhibition = (Average DS for Control -Treatment DS)/(Average DS for Control) x 100%. Where treatment DS is the disease severity for each leaf except those of the control. Plots were generated using the 'ggplot2' package in R version 4.3.2.

## Supporting information

**S1 Fig. Mass spectrometry confirms identical molecular weights for the products in NCR13_peak 1 and NCR13_peak 2. (A-B)** The mass spectra of NCR13_peak 1 and NCR13_peak 2 shows major peaks corresponding to different charge states of the peptide. MW = Molecular weight.
(TIF)

**S2 Fig. Identification of the disulfide bond pattern for NCR13_PFV2 using mass spectrometry.** The primary amino sequence of NCR13 contains six cysteine residues which means there are 15 possible combinations of disulfide bond formation. In nature, each NCR is expected to fold with one specific disulfide bond combination. High-resolution mass spectrometry on a trypsin digested sample unambiguously shows the pattern is C4-C10, C15-C30, and C23-C28 for NCR13_PFV2 as illustrated in the analysis above. There are at least three major digestion products that correspond to a C15-C30 disulfide and two corresponding to a C23-C28 disulfide. This information was crucial in solving the NMR solution structure for NCR13_PFV2.
(TIF)

**S3 Fig. NCR13_PFV1 and NCR13_PFV2 fold differently.** Cartoon representation of the structures of NCR13_PFV1 (purple) and NCR13_PFV2 (grey) superimposed on the β-sheet (V20 –V31). Helices are illustrated as cylinders. In NCR13_PFV2, the helix sits off to the side of the *β-sheet* while in NCR13_PFV1, it sits over the top of the β-sheet's face. This results in a slightly more compact structure for NCR13_PFV1.
(TIF)

**S4 Fig. Comparison of the experimental structure of NCR13_PFV1 with a structure calculated with a reversed disulfide bond pattern and with the predicted AlphaFold structure.**

 

(A) Cartoon representation of the structures of NCR13_PFV1 (purple) and a NCR13_PFV1 structure calculated with a reverse disulfide bond pattern (yellow) superimposed on the β-sheet (V20 –V31). Helices are illustrated as cylinders with the position of C4 colored red in both structures. Given that the chemical shift data shows that both NCR13_PFV1 and NCR13_PFV2 contain a C15-C30 disulfide bond, there are two possible disulfide bond patterns possible for NCR13_PFV1: C4-C23/C10-28 (purple) and C4-C28/C10-C23 (yellow). Long range experimental NOEs between the N-terminal region of NCR13_PFV1 (T1-P3) and the β-strand were satisfied with the former disulfide pattern. In the reversed, C4-C28/C10-C23 disulfide pattern, the direction of α1 is also reversed relative to the β-strand and this prevents the N-terminal region (T1-P3) from making close contact with the β-strand. (B) Cartoon representation of the structures of NCR13_PFV1 (purple) and the AlphFold predicted structure (green) superimposed on the β-sheet (V20 –V31). Helices are illustrated as cylinders with the position of C4 colored red in both structures. While the *β*-sheet is similar in both the experimental and predicted structures, AlphaFold predicted a slightly shorter α-helix displaced two residues (D7-K11 versus Q5-C10 experimentally) that packs against the β-strand a bit differently. Note that the N-terminal region (T1-P3) folds towards the β-strand in the AlphaFold predicted structure as observed experimentally.
(TIF)

**S5 Fig. Replacing all six cysteine residues with serine in NCR13 abolishes activity against *B*. *cinerea*.** Comparison of the antifungal activity of a full disulfide knockout, NSR13, and chemically synthesized NCR13 (NCR13_CS) Fungal cell viability assay performed with resazurin. A change from blue to pink/colorless signals resazurin reduction and indicates metabolically active *B*. *cinerea* germlings after 60 h.
(TIF)

**S6 Fig. Identification of the NCR13 active motif.** (A) Antifungal activity of synthetic NCR13 and NCR13 alanine mutant variants. The latter constructs were generated by substituting a window of alanine residues within the NCR13 core sequence, leading to the creation of constructs NCR13_AlaV1 through V5. (B) Comparison of the antifungal activity of chemically synthesized NCR13_AlaV3 and *P*. *pastoris* produced NCR13_AlaV3. All experiments were performed against *B*. *cinerea* using the resazurin fungal cell viability assay. A color change from blue to pink/colorless signals resazurin reduction indicating metabolically active fungal spores after 48 h. For each concentration of peptide, three biological replicates were used. Calculated MIC values are provided on the side of each experiment.
(TIF)

**S7 Fig. Modest differences in the surface charge distribution between NCR13_PFV1 and NCR13_PFV2.** Electrostatic potentials at the solvent-accessible surface of NCR13_PFV1 and NCR13_PFV2 with negative regions colored red and positive regions colored blue. All four faces of the protein are shown by rotating the first structure ~90˚ around the y-axis (counterclockwise looking down from the top of the y-axis). Each peptide contains ten positively charged and two negatively charged side chains that are all solvent exposed. The most significant difference in the distribution of these charged residues is that they are primarily clustered in one region in NCR13_PFV2, while in NCR13_PFV1, they are generally separated into two regions by a row of hydrophobic (F27, F13, and A14) or neutral (C28) amino acids. Perhaps this different distribution in charge contributes to the different antifungal properties of the two peptides.
(TIF)

**S8 Fig. Peptide oligomerization in the presence of PI(4,5)P2.** To determine if the peptide oligomerized in the presence of lipid, chemical crosslinking experiments were performed with NCR13_PFV1 and NCR13_PFV2 in the presence of different concentrations of PI(4,5)P2. Following incubation with the biochemical cross-linker bis(sulfosuccinimidyl)suberate (BS3), the peptide was run on an SDS-PAGE gel and the product visualized by silver staining. MW indicates the lane with molecular weight protein marker. The shown image is representative of three independent experiments.
(TIF)

**S9 Fig. NCR13 antibodies exhibit similar binding affinity to both disulfide variants, PFV1 and PFV2.** Dot blot analysis of purified NCR13_PFV1 and PFV2 using anti-NCR13 antibody (0.1 μg/mL) followed by goat anti-rabbit IgG HRP (Cytiva RPN4301) at 1:20,000 dilution.
(TIF)

**S10 Fig. NCR13_PFV2 exhibits rRNA binding at higher concentrations.** Electrophoretic mobility shift assay (EMSA) to determine if NCR13_PFV1 and NCR13_PFV2 bind rRNA. *B. cinerea* 28s and 18s rRNA was used to assess binding by electrophoresis on an agarose gel (1%). Peptide concentrations are indicated above the lanes. The first lane on the left contains molecular weight markers.
(TIF)

**S11 Fig.  Semi-*in planta* antifungal activity of NCR13_PFV1 and NCR13_PFV2 against *B. cinerea* on detached pepper leaves** (A) Model of each treatment location on the pepper leaf (B) Representative pictures (under white light and with CropReporter) showing the antifungal activity of NCR13_PFV1 and NCR13_PFV2 at 3, 1.5, and 0.09 μM against *B. cinerea* on detached pepper leaves. N = 4, N refers to biological replicates. (C) Photosynthetic efficiency (Fv/Fm) measurements of diseased lesions. In the box plot, horizontal lines represent the median and boxes indicate the 25th and 75th percentiles. Statistical significance between control and treated samples was tested using One way ANOVA with Dunnett Multiple comparison test.
(TIF)

**S12 Fig. *In planta* curative antifungal activity of NCR13_PFV1 and NCR13_PFV2 against *B. cinerea* on tomato plants.** Four-weeks old tomato plants sprayed with 1 mL of a $5 \times 10^4$ *B. cinerea* spore suspension followed 24 h later by a spray containing either 2 mL of water or NCR13_PV1 or NCR13_PV2 at 3 μM. (A) Representative pictures showing the curative antifungal activity of NCR13_PFV1 and NCR13_PFV2 at 3 μM against *B. cinerea* on tomato leaves. (B) Disease severity %; (C) Disease inhibition %. For panel (C) and (D) each data point represent mean ± SEM denoted by a dot. The average of 5 leaves per plant for 4 plants per treatment. Statistical significance between control and treated samples were tested using One way ANOVA with a Dunnett Multiple comparison test. Three independent experiments were conducted with similar results.
(TIF)

**S1 Table. Yields of NCR13 from HPLC peaks 1 and 2 obtained from various purification batches of from *Pichia pastoris*.**
(PDF)

**S2 Table. Chemical shifts (ppm) for the $^{13}C^{\beta}$ carbons of the six cysteine residues in oxidized NCR13_PFV1 and NCR13_PFV2.**
(PDF)

**S3 Table. Summary of the structural statistics for the solution structures of NCR13_PFV1 and NCR13_PFV2.**
(PDF)

**S4 Table. Summary of the molecular weight of the NCR13 disulfide knockout variants determined using mass spectrometry.**
(PDF)

## Acknowledgments

The structures for NCR13 were solved using resources at the W.R. Wiley Environmental Molecular Sciences Laboratory, a national scientific user facility sponsored by U.S. Department of Energy's Office of Biological and Environmental Research (BER) program located at Pacific Northwest National Laboratory (PNNL). Battelle operates PNNL for the U.S. Department of Energy. We also acknowledge imaging support from the Advanced Bioimaging Laboratory (RRID:SCR_018951) at the Donald Danforth Plant Science Center (DDPSC) and usage of the Leica SPX-8 acquired through an NSF Major Research Instrumentation grant (DBI-1337680). We acknowledge the DDPSC Proteomics & Mass Spectrometry Facility and supported by the National Science Foundation under Grant No. DBI-1827534 for acquisition of the Orbitrap Fusion Lumos LC-MS/MS. The greenhouse and growth chamber space provided by the DDPSC Plant Growth Facility is acknowledged. We also thank Ruby Tiwari for providing her comments on this manuscript.

## Author Contributions

**Conceptualization:** Kirk J. Czymmek, Dilip M. Shah.

**Data curation:** James Godwin, Arnaud Thierry Djami-Tchatchou, Siva L. S. Velivelli, Mowei Zhou, Garry W. Buchko.

**Formal analysis:** James Godwin, Arnaud Thierry Djami-Tchatchou, Siva L. S. Velivelli, Meenakshi Tetorya, Raviraj Kalunke, Mowei Zhou, Garry W. Buchko.

**Funding acquisition:** Kirk J. Czymmek, Dilip M. Shah.

**Investigation:** James Godwin, Arnaud Thierry Djami-Tchatchou, Siva L. S. Velivelli, Meenakshi Tetorya, Raviraj Kalunke, Ambika Pokhrel, Mowei Zhou, Garry W. Buchko.

**Methodology:** James Godwin, Arnaud Thierry Djami-Tchatchou, Siva L. S. Velivelli, Ambika Pokhrel.

**Project administration:** Garry W. Buchko, Kirk J. Czymmek, Dilip M. Shah.

**Resources:** Raviraj Kalunke, Kirk J. Czymmek, Dilip M. Shah.

**Supervision:** Kirk J. Czymmek, Dilip M. Shah.

**Validation:** James Godwin, Raviraj Kalunke, Mowei Zhou, Garry W. Buchko, Kirk J. Czymmek, Dilip M. Shah.

**Visualization:** James Godwin, Arnaud Thierry Djami-Tchatchou, Meenakshi Tetorya, Mowei Zhou, Garry W. Buchko.

**Writing – original draft:** James Godwin, Dilip M. Shah.

**Writing – review & editing:** James Godwin, Arnaud Thierry Djami-Tchatchou, Siva L. S. Velivelli, Meenakshi Tetorya, Raviraj Kalunke, Ambika Pokhrel, Mowei Zhou, Garry W. Buchko, Kirk J. Czymmek, Dilip M. Shah.

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
