## [Decision Letter · Decision Letter 0]

21 Oct 2024

Dear Dr. Shah,

Thank you very much for submitting your manuscript "Chickpea NCR13 disulfide cross-linking variants exhibit profound differences in antifungal activity and modes of action" for consideration at PLOS Pathogens. As with all papers reviewed by the journal, your manuscript was reviewed by members of the editorial board and by several independent reviewers. The reviewers appreciated the attention to an important topic. Based on the reviews, we are likely to accept this manuscript for publication, providing that you modify the manuscript according to the review recommendations.

Sincerely,

Savithramma P. Dinesh-Kumar

Section Editor

PLOS Pathogens

Savithramma Dinesh-Kumar

Section Editor

PLOS Pathogens

Michael Malim

Editor-in-Chief

PLOS Pathogens

orcid.org/0000-0002-7699-2064

Reviewer Comments (if any, and for reference):

Reviewer's Responses to Questions

**Part I - Summary**

Reviewer #1: The manuscript by Godwin et al focuses on the biochemical and functional characterization of the nodule-specific cysteine-rich peptide 13 (NCR13) from chickpea. NCRs are small secreted cationic peptides with antimicrobial activity, produced by legumes of the inverted-repeat lacking clade. They typically contain four or six conserved cysteine (Cys) residues, the oxidization of which leads to the formation of intramolecular disulfide bonds between them, to enhance the stability of the peptides. However, the pattern of cross-linking between these Cys residues via disulfide bonds is not necessarily fixed for individual peptides but may vary, thereby giving rise to peptides that differ in their tertiary structure and consequently biologically activity.

In this study the authors compared the biochemical characteristics and mode of action (MOA) of the two peptide variants (i.e. NCR13_PVF1 and NCR13_PVF2) produced by NCR13 when expressed in Pichia pastoris. The authors show by mass spectrometry of the trypsin-digested peptides (for NCR13_PFV2) and AlphaFold predictions (NCR13_PFV1) that the two peptides differ in their disulfide cross-linking pattern, which based on an NMR structural analysis resulted in different tertiary structures as well. Subsequent analysis showed that NCR13_PVF1 and NCR13_PVF2 also differ in their biological activities in terms of ability to entry and resist proteolytic degradation in fungal cells, interact with rRNA and inhibit protein translation, and inhibit fungal growth under in vitro and in plant conditions.

The manuscript is in general clearly written and logically structured. Experiments are for the most part complete and rigorous, contain the right controls, and the data that were produced are adequately presented in figures and tables. Conclusions and postulates made by the authors are also fairly supported by the data. Overall, this is a nice study that enriches our understanding of the MOA of cysteine-rich antimicrobial peptides. However, I would appreciate it if the authors could clarify and discuss the following points in their manuscript.

Reviewer #2: This is a very in-depth analysis of the antifungal activities of NCR13 from a legume species. I learned a lot by reading this manuscript. I have the following questions that I hope the authors can enlighten me:

**Part II – Major Issues: Key Experiments Required for Acceptance**

Reviewer #1: 1) How do the different structural arrangements adopted by NCR13_PFV1 and NCR13_PFV13 affect the charge distribution on their surfaces and their potential interaction with anionic phospholipids present on fungal membranes? How are amphipathicity or hydrophobicity compare in these structures and to what extend differences in these properties might have affected the results? Please discuss in the manuscript, including possible consequences on the peptides’ antifungal activities.

2) Along the same lines, the authors have utilized alanine scanning variants to determine the sequence motif governing the antifungal activity of NCR13. However, what are the differences in cationicity, amphipathicity and hydrophobicity among these variants and can differences in these properties be the reason behind the loss activity in some variants? The authors should present such changes in protein properties and discuss how their results might have been affected or dictated by these changes.

3) To what extent is the antifungal activity of NCR13 variants charge-driven and determined by electrostatic attraction with the negatively charged fungal membrane? How is the activity of the peptides affected by increasing salt concentrations? What is the salt concentration the experiments have been performed? Please discuss and if possible, test as well.

4) The NCR13 construct with all six Cys residues replaced with Ser (NSR13) has no activity against B. cinerea, which in part has led the authors to conclude that disulfide bonds are essential to the biological activity of NCR13 (lines 318-322). However, the chemically synthesized NCR13 which lacks disulfide bonds as well has a MIC of 3 uM against B. cinerea (line 279). Why this difference between the two isoforms?

5) Functional analysis of the disulfide-bonds in NCR13 by means of site-directed Cys to Ser mutagenesis led the authors to conclude that “…the disulfide pairing of C4-C23 and C15-C30 is essential for the formation of NCR13_PFVs” (lines 310-311). However, NCR13-PVF2 lacks the C4-C23 and it is still functional. Please modify or further clarify your conclusion statement.

6) The authors observed that NCR13_PFV1 can permeabilize the plasma membrane of B. cinerea faster and at lower concentrations as compared to NCR13_PFV2. An explanation given by the authors is that NCR13_PFV1 adopts a more compact which enables it to cross more readily the fungal cell wall and membrane (lines 633-635). This argument assumes that the two variants are utilizing the same mechanism to cross the membranes and enter the cytosol, but what are the indications for that? Given their localization in different cellular foci, can it be that one, or both but at different degrees depending on properties and peptide concentration, utilizes an energy-dependent endocytic pathway as opposed to an energy independent mechanism? Please discuss in the manuscript.

Reviewer #2: None. I would like to see some additional data (please see the next section), but they are not major issues.

**Part III – Minor Issues: Editorial and Data Presentation Modifications**

Reviewer #1: Line 35: It is indicated in the Abstract that the net charge of NCR13 is +9, but in Results is reported as +8. Which one is correct?

Line 48: change to “…a structural framework for designing…”

Line 57: add ‘of’ prior to ‘legume’.

Line 79: add ‘the’ prior to ‘evolution’.

Line 122: it is mentioned that “NCR13 disulfide variants displayed notable differences in fungal cell entry, cell permeabilization,..”. is there a difference between the two? (cell entry and cel permeabilization).

Reviewer #2: 1. It is believed that NCR peptides evolved to interact with rhizobia. Could the authors speculate why NCR13 has such high activities against several fungi? Is NCR13 special in that it has evolved to target fungi? Or does it have functions similar to other NCR peptides in the legume-rhizobia symbiosis, but such activities happen to be potent on fungi as well?

2. Related to the above question, is the behavior of NCR13_PFV1 observed in this study relevant to its function in the legume-rhizobia symbiosis? Specifically, the authors showed that this peptide localizes to the nucleoli, binds rRNA, and inhibits translation in fungi. The bacterial counterparts are either very different, or missing. Do the authors believe NCR13 interferes with protein translation in rhizobia also? If so, does it act through a conserved mechanism?

3. Can the authors speculate which form is present in the nodule, or do they believe NCR13 exists in more than one fold, as observed in Pichia?

4. Still related to the first question, does any of the three forms of NCR13 (PFV1, PFV2, and chemically synthesized) show activities against rhizobia? If so, do they have even lower MICs when tested against rhizobia?

5. If the peptide is highly toxic to three very different pathogenic fungi, how could the authors produce this peptide in Pichia, another fungal species? Is it toxic to Pichia, and did the author have to adjust the induction condition so as to be able to isolate the peptide before the yeast cells died?

6. Perhaps related to the above question, the authors noted that the PFV1 form, which is much more active, consistently yielded less than PFV2. Is it because the PFV1 form is more toxic to Pichia and thus cannot be produced in large quantities?

7. The chemically synthesized NCR13 has a retention time different from either PFV1 or PFV2. How do the authors think it is folded? Is it reduced? Or is it simply misfolded? Is it a cautionary tale for studies that use chemically synthesized NCR peptides in general?

PLOS authors have the option to publish the peer review history of their article (what does this mean?). If published, this will include your full peer review and any attached files.

Reviewer #1: No

Reviewer #2: No

Figure Files:

Data Requirements:

Reproducibility:

References:

---

## [Editor Report · Decision Letter 1]

11 Nov 2024

Dear Dr. Shah,

We are pleased to inform you that your manuscript 'Chickpea NCR13 disulfide cross-linking variants exhibit profound differences in antifungal activity and modes of action' has been provisionally accepted for publication in PLOS Pathogens.

Best regards,

Savithramma P. Dinesh-Kumar

Section Editor

PLOS Pathogens

Savithramma Dinesh-Kumar

Section Editor

PLOS Pathogens

Michael Malim

Editor-in-Chief

PLOS Pathogens

orcid.org/0000-0002-7699-2064
---

## [Editor Report · Acceptance letter]

19 Nov 2024

Dear Dr. Shah,

We are delighted to inform you that your manuscript, "Chickpea NCR13 disulfide cross-linking variants exhibit profound differences in antifungal activity and modes of action," has been formally accepted for publication in PLOS Pathogens.

Best regards,

Michael Malim

Editor-in-Chief

PLOS Pathogens

orcid.org/0000-0002-7699-2064